# AI-driven approaches for dysgraphia diagnosis using online and offline handwriting data: A comprehensive scoping review

Avisa Fallah, Yazdan ZandiyeVakili, Hedieh Sajedi◉*

School of Mathematics, Statistics and Computer Science, College of Science, University of Tehran, Tehran, Iran

* hhsajedi@ut.ac.ir

## Abstract

Dysgraphia, a neurodevelopmental disorder impacting writing abilities, is often overlooked or misdiagnosed, affecting academic and daily life activities. It affects both children, especially school-aged, and adults, potentially stemming from brain trauma or neurological diseases. Early diagnosis is crucial for effective management and intervention. With advancements in artificial intelligence (AI), there is potential to improve early diagnosis and intervention through automated detection methods. This scoping review aims to explore available studies employing AI-based models for dysgraphia diagnosis and identify the highest performance models and their challenges to suggest future improvements. Comprehensive searches were conducted across PubMed, Scopus, Web of Science, IEEE Xplore, and SpringerLink until April 2024. Included studies are original research articles focusing on dysgraphia detection using AI-based predictive models with image processing techniques on handwritten images. Data extraction was done using a structured form in Google Sheets, capturing study characteristics, datasets, methods, and results. Out of 177 initial papers, 21 met the inclusion criteria. Studies span nearly a decade, with 38% being conference papers and 62% journal articles from various countries. AI models, particularly Convolutional Neural Networks (CNNs) and Support Vector Machines (SVMs), demonstrated high accuracy rates, often surpassing 90%, with comprehensive feature extraction methods enhancing performance. Significant challenges include small sample sizes and language-specific models. AI-based models, especially advanced ones like CNNs and SVMs, significantly enhance dysgraphia diagnosis, offering faster and more accurate assessments than traditional methods. Future research should focus on larger, more diverse datasets, language-independent models, and standardized evaluation metrics. Integrating AI-based tools in educational and healthcare settings can revolutionize dysgraphia management, improving academic outcomes and support for affected individuals.

**Data availability statement:** The data supporting this review's findings include publicly available and private datasets and articles from various sources. The datasets and papers reviewed in this study are specified in the respective sections of the paper. Public datasets include: • Dysgraphia Detection Through Machine Learning: (https://github.com/peet292929/Dysgraphia-detection-through-machine-learning) • Dysgraphia Handwriting Sample: Contact Association of Dyslexia Malaysia (ADM) • IAM Handwriting Database: (https://fki.tic.heia-fr.ch/databases/iam-handwriting-database) • Handwritten Digits: (https://ijece.iaescore.com/index.php/IJECE) The paper references these sources, and access to these materials is provided through the respective links. Access to additional papers and datasets mentioned in this review may require institutional access or a subscription. Additionally, any summary tables, figures, or supplementary materials created during the preparation of this review are included as Supporting information files with this article.

**Funding:** The author(s) received no specific funding for this work.

**Competing interests:** The authors have declared that no competing interests exist.

## 1. Introduction

Writing, a crucial skill developed early in life, is linked to overall achievement, particularly in academic pursuits, and can take up to fifty percent of the school day [1,2]. Poor handwriting can lead to lower self-esteem and social difficulties, with students frequently misunderstood as careless or unmotivated instead of identified as having a learning disability [3–5]. Dysgraphia (Fig 1) is a neurodevelopmental disorder predominantly affecting boys, impairing writing abilities from childhood through adulthood [6,7]. It may arise from brain trauma, infections, or diseases that harm the parietal lobe. It is commonly observed in conditions like cerebrovascular injury, Huntington's disease, multiple sclerosis, Alzheimer's disease, vascular dementia, and Parkinson's disease [8]. Symptoms include problems with letter formation, legibility, spacing, spelling, and grammar that do not align with the individual's cognitive levels despite appropriate learning [7,8].

Recent research suggests that various subtypes of dysgraphia may be linked to distinct neurological mechanisms [9], with significant interrelations among brain regions governing routine processes, linguistic abilities, and movement coordination, particularly the cerebellum [8]. Functional imaging and case studies indicate that cerebellar damage can lead to acquired dysgraphia symptoms, underscoring its role in language and automaticity, especially in coordinating writing [10,11]. The cerebellum may help develop a neural system critical for writing, disruption of which could cause various functional impairments [12]. Additionally, genetic studies highlight a hereditary component in verbal executive functions, orthographic skills, and spelling abilities [13–15]. Notably, genes on chromosomes 15 and 6 have been linked to difficulties in reading, spelling, and phonemic awareness, respectively [16,17].

Often misdiagnosed or overlooked, dysgraphia frequently coexists with other learning and psychiatric disorders and can have lifelong impacts and hinder vocational progress and daily activities in adults [18]. Effective management of dysgraphia, crucially involving the educational system, is essential for proper diagnosis and treatment.

Dysgraphia diagnosis requires a team effort from specialists in education, psychology, and medicine to evaluate a student's handwriting abilities and discern contributing factors to their writing challenges [20]. Before a thorough evaluation, it is crucial to rule out other conditions that could affect handwriting, such as hearing loss, visual impairments, or inadequate training [21]. Key assessment areas include writing speed, clarity, spelling consistency, and pencil grip to diagnose dysgraphia effectively [22,23]. Although a universally standardized medical assessment method for diagnosing dysgraphia does not yet exist, recent advances in artificial intelligence suggest that machine learning (ML) could be a viable solution for early detection [5,22,24]. These methods could offer faster and more accurate evaluations than traditional assessments by experts, which are subject to human biases and the variability of specialist expertise. Additionally, manual assessments demand considerable time and human resources.

**Fig 1. Sample handwritten letter from a patient diagnosed with dysgraphia [19].**

In our scoping review, we aim to identify and explore valuable research that employs computational approaches through AI-based models for dysgraphia detection in handwriting images. As this area of research is relatively unexplored and lacks in-depth discussion, this review aims to identify available studies, collect related information, especially the datasets used, crucial features, and challenges of building these models, and employ a variety of methods and study designs to address broad and complex questions, rather than focusing on specific outcomes. We will also discuss future trends in dysgraphia detection models based on handwriting and other methods for recognizing dysgraphia beyond handwriting images. Thus, we conducted this scoping review instead of a systematic review.

### 1.1. Objectives

1  Identify available studies that used AI-based models for dysgraphia diagnosis and explore the current dominant use of AI in expediting dysgraphia detection.

2  Explore the challenges of building computational AI-based models under current conditions and the limitations of each study.

3  Determine the best models with high accuracy and performance in dysgraphia diagnosis, examine their effectiveness in dysgraphia detection, and consider possible future improvements in these models.

4  Examine the integration of machine learning techniques with traditional diagnostic methods.

5  Analyze the training datasets used for developing AI models in dysgraphia diagnosis.

6  Review technological advancements and their implications for future AI models in dysgraphia diagnosis.

## 2. Methods

The primary aim of this scoping review is to investigate, identify, and evaluate mainstream AI-based models used to detect dysgraphia through the analysis of handwritten images. Based on our aim, a scoping review focusing on leading and groundbreaking models that have emerged over the past decade was undertaken, aligning with the guidelines reported in the Scoping Reviews (PRISMA-ScR) Statement [25].

### 2.1. Eligibility criteria

All original research articles published until April 2024 that explore dysgraphia detection using predictive AI-based models with image processing techniques on handwritten images were included. This criterion ensures a focused review of

advanced AI methodologies for diagnosing dysgraphia. Papers were excluded if they did not use AI-based predictive models for dysgraphia detection or relied on other data sources like EEG or brain scans. Additionally, review articles, studies on other learning disabilities (e.g., dyslexia, dyscalculia), gray literature, and conference abstracts were excluded to maintain a clear focus.

## 2.2. Information sources

To comprehensively gather relevant studies, we searched five major scientific databases until April 2024: PubMed, Scopus, Web of Science, IEEE Xplore, and SpringerLink. Team members initially drafted the search strategies and then refined them under the guidance of our supervisor. The final search outcomes were exported to EndNote, and the team carefully removed duplicate entries through a double-check process.

## 2.3. Search

The search strategy was developed using a combination of keywords organized into three categories: Model, Technique, and Disease (Table 1). For the disease category, which focuses on Dysgraphia, we also included relevant *MeSH* [26] terms to ensure comprehensive coverage. We employed the 'AND' operator between different categories and the 'OR' operator within the same category to effectively refine and broaden our search results as needed.

## 2.4. Selection of sources of evidence

To ensure alignment with the defined eligibility criteria and maintain consistency among reviewers, all titles and abstracts of publications were double-screened by two reviewers. In cases of unclear points, both reviewers screened the full text. Any disagreements regarding inclusion were resolved through discussions with the supervisor until a consensus was achieved. The same two reviewers conducted full-text screening to extract the features. All data were managed using a spreadsheet environment.

## 2.5. Data charting process

The data charting process was meticulously conducted using a structured form developed in Google Sheets. This form included columns for key study characteristics, datasets, methods, and results. The selection of items for charting was based on previously established eligibility criteria and reviewed by two reviewers. One of the reviewers participated in the data charting process, while the other independently charted each study. Discrepancies were resolved through thorough discussions. Inconsistencies were addressed during team meetings, and data verification involved confirming details with original study authors when necessary. The charting process was iterative, with the form updated based on initial findings. Additional variables were added after the first round of charting to ensure a comprehensive data collection. Data confirmation was supervised by the supervisor, ensuring accuracy and reliability.

**Table 1. List of keywords applied in the search strategy.**

| Model-Related Words | Technique-Related Words | Disease-Related Words |
|---|---|---|
| AI/Artificial Intelligence/ML/ Machine Learning/ DL/Deep Learning/ Image Processing | CNN/Convolution Neural Networks/SVM/Support Vector Machine/NNs/Neural Networks/AdaBoost/Adaptive Boosting/ U-Net/UNet/VGGNet/ResNet/Residual Net/Random Forest/ Decision Tree/ PCA/LDA/AutoEncoder/GAN/Generative Adversarial Network/GoogleNet/MobileNet/DenseNet/YOLO/ RetinaNet/Naïve Bayes/MLP/Multi-Layer Perceptron | Dysgraphia/ **Dysgraphias**/ **Argraphia**/ **Argraphias** |

**\*Mesh terms are bold**

## 2.6. Data items

The review included qualitative and quantitative data, capturing a comprehensive spectrum of study characteristics and outcomes. Quantitative data, such as accuracy rates, were charted alongside qualitative descriptions of study objectives. Interpretation was necessary for categorizing intervention types and understanding contextual factors from study descriptions, particularly for variables like feature extraction methods. The final charting form comprised columns for study characteristics (e.g., author, year), datasets used, methods (e.g., model training, evaluation metrics), and results (e.g., accuracy, precision, strengths). Variables collected included publisher details, dataset features, size, methods of data collection, detailed methodological approaches, and performance metrics. For instance, study characteristics captured information such as publisher impact factor and open access status, while dataset details included features like X/Y coordinates and pen position. This comprehensive approach ensured a thorough synthesis of the available evidence, providing valuable insights into detecting dysgraphia through various methodologies.

## 2.7. Critical appraisal of individual sources of evidence

In our study, we critically appraised the included studies to guarantee the quality and relevance of the evidence. This appraisal was carried out using the Critical Appraisal Skills Programme (CASP) [27] checklists for qualitative studies and the Newcastle-Ottawa Scale [28] for quantitative studies. Each study was evaluated based on its methodological rigor, clarity of reporting, and relevance to the research question, focusing on dysgraphia diagnosis using handwritten images.

The CASP checklist facilitated a structured review of qualitative studies, assessing aspects such as the validity of the research, clarity in reporting, and appropriateness of the methodology. Studies were scored on key components such as the research aim, design, recruitment strategy, data collection methods, and findings.

For quantitative studies, the Newcastle-Ottawa Scale was employed to evaluate the quality of non-randomized studies. It specifically focused on three main criteria: selection of study groups, comparability of groups, and the ascertainment of either the exposure or outcome of interest. Studies were awarded stars for meeting specific criteria within these categories, with a maximum of nine stars indicating the highest quality.

### 2.7.1. Proposed appraisal method explanation.
In this study, we implemented a detailed scoring system to critically appraise the included studies, ensuring a rigorous evaluation of their quality and relevance. This system is based on three main criteria: validity, results, and relevance, with each criterion contributing to an overall score that reflects the study's robustness and applicability.

a) **Validity** was assessed according to the variety of models applied in the study. Studies that applied more than four models received the highest score of 3 points, those using 3 or 4 models received 2 points, and studies applying fewer than three models received 1 point. This measure reflects the comprehensiveness of the study's approach to model evaluation and comparison, which is crucial for robust and reliable findings.

b) **Results** were evaluated by examining the reported accuracy of the study's outcomes. This measure acknowledges the significance of precise and reliable results in validating a study's conclusions. Studies reporting an accuracy greater than 90% were awarded 3 points, those between 80% and 90% received 2 points, and studies with accuracy between 70% and 80% received 1 point. Studies reporting accuracy below 70% or not reporting accuracy at all received 0 points.

c) **Relevance** was determined by the extent to which the study focused on dysgraphia, handwriting, and classification problems. Studies mentioning "dysgraphia" and "handwriting" received the highest score of 3 points, indicating a direct relevance to our research focus. Studies mentioning only "dysgraphia" received 2 points, while those mentioning only "handwriting" received 1 point. Additionally, studies that addressed a classification problem received 1 point. Studies that did not mention either term or address classification problems received 0 points.

The total score for each study, ranging from 0 to 9, was calculated by summing the points from all three criteria. This total score allowed us to classify the studies into categories of high quality (7–9 points), moderate quality (4–6 points), and low quality (0–3 points). This structured approach ensured a comprehensive and transparent appraisal of the evidence, allowing us to base our conclusions on the most reliable and relevant studies available.

## 3. Results

### 3.1. Selection of sources of evidence

The initial search results were collected and consolidated using EndNote [29], with duplicates removed (Fig 2). The screening process and data selection, as outlined in Fig 2, were based on PRISMA guidelines [30]. From an initial extraction of 177 papers, 135 remained after duplicate removal. Abstracts were reviewed to exclude studies not primarily focused on dysgraphia, such as those diagnosing Alzheimer's and Parkinson's disease through handwritten images or those that were review papers. Additionally, studies employing diagnostic methods like EEG or brain scans were set aside, keeping only those involving handwritten images. Applying these inclusion and exclusion criteria reduced the number of relevant papers to 37. The review process advanced with a thorough examination of the complete texts, further excluding those that were not primarily related to dysgraphia or did not conform to models based on handwritten images, resulting in 21 retained papers. This study is a scoping review focused on identifying papers that diagnose dysgraphia using computational methods.

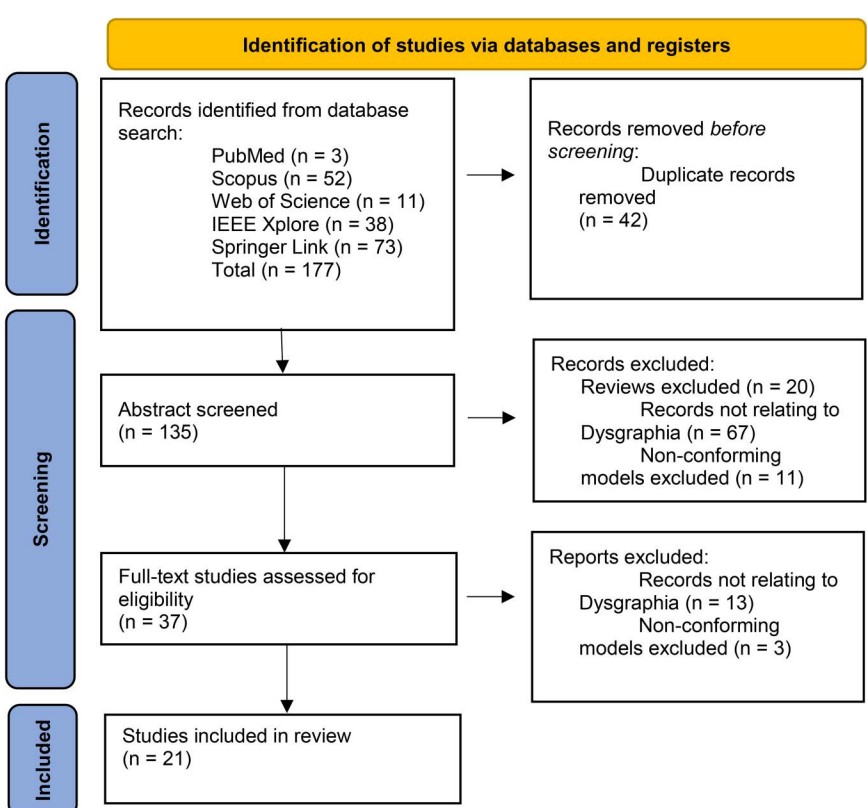

**Fig 2. Overview of study selection using the PRISMA framework.**

## 3.2. Literature context and research gaps

The publication dates span nearly a decade, from 2015 to the present (Fig 3). Of the 21 selected papers, 8 (n = 8, 38%) [31–37] are conference papers, and the remaining 13 (n = 13, 62%) [20,38–57] are journal articles. The papers originate from various countries, including India, Qatar, Spain, Switzerland, Slovakia, the Czech Republic, Israel, France, the USA, Malaysia, Morocco, and Austria. Additionally, some papers are collaborative efforts involving researchers from multiple

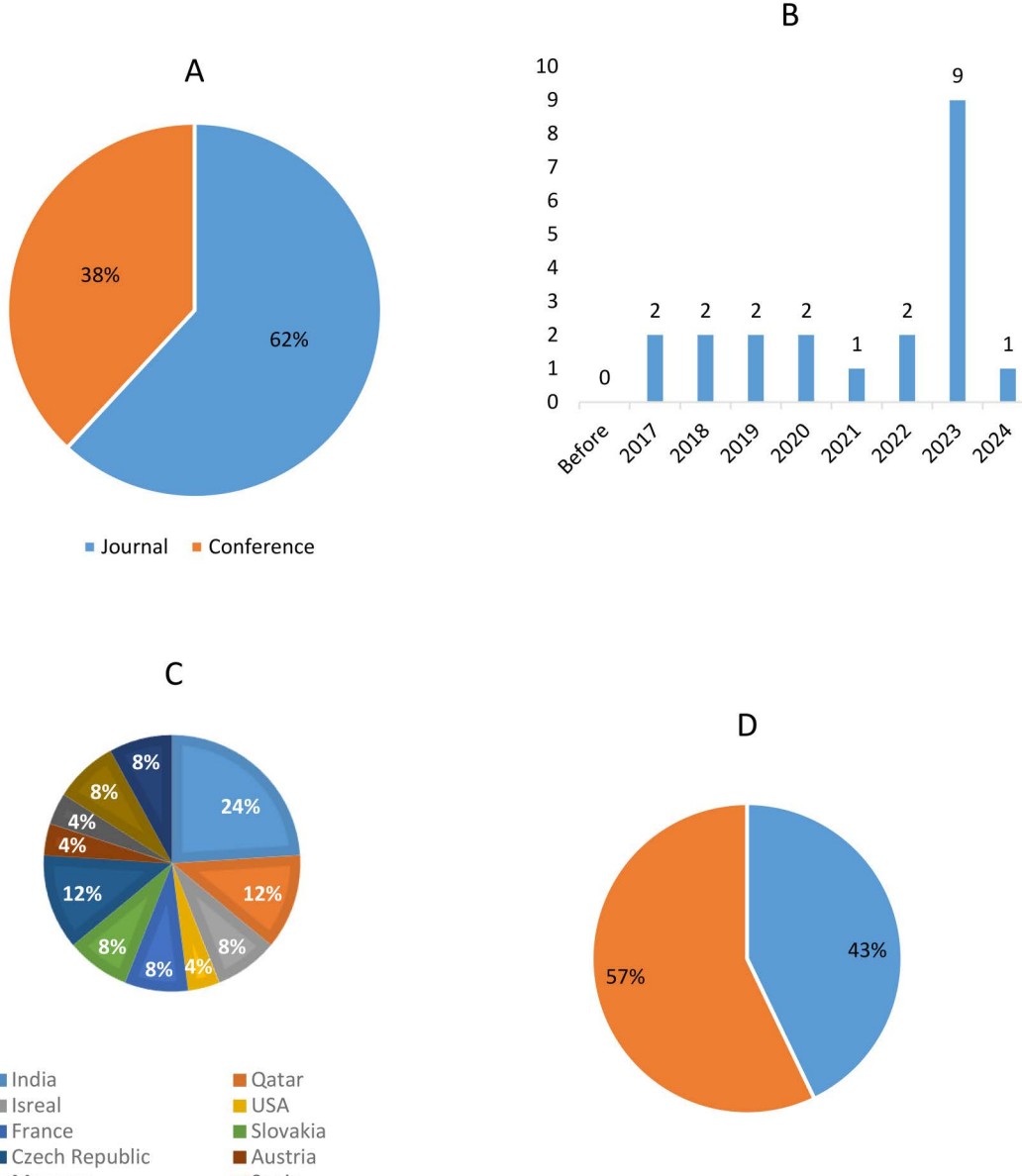

**Fig 3. General characteristics of the included studies.** A. Distribution of journal articles versus conference proceedings; B. The number of publications by year shows the publication trends from 2015 to 2024; C. Geographic distribution of studies by country; D. Proportion of public versus private access to studies.

nationalities. In addition, only nine of the included studies are open-access (n = 9, 42%) [20,38–40,43–45,48], and the 12 other papers are private (n = 12, 58%) [31–33,41,42,34–37,46–47,58].

While our review provides detailed insights into the current AI methodologies for diagnosing dysgraphia, it is essential to contextualize this within the broader academic dialogue. Previous reviews have provided foundational knowledge in this field; however, most adopt a clinical perspective, prioritizing diagnostic criteria and behavioral assessments over AI's potential to transform dysgraphia diagnosis [49,50]. Moreover, few are dedicated solely to dysgraphia, often embedding it within broader discussions of specific learning disabilities (SLDs) like dyslexia and dyscalculia, thus diluting focus on dysgraphia-specific challenges [51–53]. Even reviews exploring AI applications, such as [54,55], typically emphasize traditional machine learning methods (e.g., SVM, AdaBoost) and overlook state-of-the-art approaches, notably the integration of traditional and modern techniques, such as combining handwriting image analysis with dynamic features like pen pressure or motor patterns, which our review examines in detail. Additionally, many prior reviews are outdated, failing to reflect recent AI advancements. One of the more comprehensive recent works is the review by Kunhoth et al. (2024) [56], which provides a broad overview of various learning disorders and explores both machine learning and non-machine learning approaches for their prediction. While their work offers valuable general insights and effectively groups and compares different models, it lacks the depth of analysis found in our study. In contrast, our review includes more specific technical details, especially about the used datasets, considers a wider array of models, and delves into how they are implemented. It also provides deeper guidance and recommendations, making it more actionable for researchers and practitioners. Together, these additions make our review not only complementary to previous works but also a necessary update for the current state of AI research in this domain. Our work addresses these gaps by providing a dysgraphia-specific, up-to-date synthesis of AI applications, comprehensively covering both traditional and cutting-edge methods. This contribution highlights the multidisciplinary relevance of AI in dysgraphia diagnosis through handwritten image analysis, underscoring its potential impact across educational and healthcare settings. Despite growing interest, a recent slowdown in publications suggests a bottleneck in data availability, emphasizing the need for improved data collection and sharing to sustain research momentum in this critical area.

### 3.3. Specific characteristics of included papers

We extracted information regarding the dataset, algorithm and methodology, and performance results and analyzed each aspect individually.

**3.3.1. Dataset characteristics.** The analyzed data in the included papers were sourced from various repositories such as databases, registers, and health information systems (Table 2). Information about datasets was extracted in both general and specific ways. Multiple tasks were applied to gather the dataset, including copying sentences, writing digits, writing individual words, and performing the BHK test in communication with humans and robots. Two papers utilized multiple datasets (n = 2, 9%) [20,33], while only three used raw handwritten images (n = 3, 14%) [32,37,57], including words, sentences, and digits. The other studies incorporated additional features, including kinematic and temporal characteristics, and documented supplementary data captured during tasks using specialized equipment such as Wacom Tablets. Open-access datasets are provided in Table 3.

**3.3.2. Studies' methodology characteristics.** Various models have been applied in these studies (Table 4), including SVM and Random Forest (RF), which were used as benchmarks to compare with the innovative model in a considerable number of studies (SVM = 11, 52% - RF = 9, 42%). One study deviated from the norm by employing a clustering method instead of a traditional classifier, utilizing it as a classifier (n = 1, 4%). The primary evaluation metric is accuracy used by 19 studies (n = 19, 90%), ranging between 79% and 99%. Additionally, three studies used ROC and AUC metrics (n = 3, 14%), and three employed the Matthews correlation coefficient (n = 3, 14%).

**Table 2. Summary of dataset information.**

| Study | Task | Features | Age | Language | Labels | Access | Participants | Total size |
|---|---|---|---|---|---|---|---|---|
| [20,31,38–40] | Letters & Sentences: varying speed | Spatial/Temporal/Dynamic | 15-Aug | Slovak | Dysgraphia/Normal | Open | 120: 57D | 480 |
| [32] | Letters | Spatial/Dynamic | – | – | Dysgraphia/Normal | Open | – | 267930 |
| [41] | – | Spatial/Temporal/Dynamic | – | Slovak | Dysgraphia/Normal | Close | – | – |
| [58] | Digits & Letters & Other | Kinematic/Error Analysis | School-going children | Sinhala | 5 Series of labels | Open | 657 | 13353 Sentences & 115320 Words |
| [33] | Letters | Kinematic/Quality/Shape Features | – | French | Normal | Open | – | 810000 characters |
| [33] | – | Kinematic/Quality/Shape Features | School-going children | French | Dysgraphia | Close | – | – |
| [42] | Think and write | Spatial/Quality/Error Analysis/Others | Grade 1–7 | – | Dysgraphia/Normal | Close | 2100 | 3131 |
| [43] | BHK Test | Spatial/Dynamic/Movement | Grade 2–5 | French | Dysgraphia/Normal | – | 580: 120D | 580 |
| [44] | BHK Test | Spatial/Dynamic/Demographic | Primary School | French | Dysgraphia/Normal | – | 448:58D | 448 |
| [34] | Copy a paragraph | Kinematic/Temporal/Spatial/Dynamic/TQWT-derived features | Grades 3 & 4 | Czech | Dysgraphia/Normal | – | 97 | – |
| [35] | Repeating Letters | Kinematic/Pen Dynamics/Others | 10–13 | Slovak | Dysgraphia/Normal | – | 78:48D | – |
| [36] | Archimedean Spiral repetitive loops/Others | Kinematic | Average of 69P and 62N | Czech | Parkinson/Normal | – | 69:33P | – |
| [45] | Copying Text | Static/Kinematic/Pressure/Tilt | Primary School | French | Dysgraphia/Normal | – | 298:56D | 298 |
| [46] | 3 Series of tasks | Temporal/Pressure/Spatial/On-Paper/Others | Grade 3 | Hebrew | Dysgraphia/Normal | – | 99:49D | – |
| [37] | Copying Text | Spatial/Other | Grade 1–7 | Kannada | Random Students | – | – | 940 |
| [47] | Letters | Kinematic/Geometric/Non-Linear Dynamic/Tablet Specific Signal/Others | Grade 3 | Hebrew | Dysgraphia/Normal | – | 54:27D | 54 |
| [48] | Letters & Words | Handwritten Images | 8-Jun | Indian | Dysgraphia/Normal | Open | 150 | – |
| [57] | Writing Digits with Interactive Robot | Handwritten Digits | 12-Jun | Latin | Dysgraphia/Normal | Open | 174 | 14004 |

**Table 3. Open access datasets.**

| Study | Dataset | Link |
|---|---|---|
| [20,31,38–40,48] | Dysgraphia Detection Through Machine Learning | https://github.com/peet292929/Dysgraph-ia-detection-through-machine-learning |
| [32] | Dysgraphia Handwriting Sample | Contact Association of Dyslexia Malaysia (ADM) [50] |
| [58] | IAM Handwriting Database | https://fki.tic.heia-fr.ch/databases/iam-handwriting-database |
| [57] | Handwritten Digits | https://ijece.iaescore.com/index.php/IJECE |

### 3.4. Results of critical appraisal within sources of evidence

The critical appraisal of the included studies revealed a range of scores spanning from 3 to 9 (Table 5, Fig 4). Three papers achieved the highest score of 9, demonstrating exceptional quality. Four papers scored 8, and five scored 7, indicating high quality. Six papers scored 6, suggesting moderate quality, while two papers scored 5. Notably, no papers received a score of 4, and one paper scored 3, reflecting lower quality. This distribution shows that most studies are of high to moderate quality, underscoring the robustness of the evidence base.

### 3.5. Comparison of mostly used predictive models

This section will discuss the most widely applied predictive models of dysgraphia detection featured in the included papers: SVM, KNN, RF, K-means, AdaBoost, and CNN. We aim to provide a comprehensive analysis by considering both their strengths and weaknesses. This balanced examination will help us understand each model's effectiveness and limitations, thereby guiding the selection of the most appropriate techniques for our research objectives.

**3.5.1. SVM.** SVM [59] is a classification model designed to identify the optimal hyperplane that separates data points in the feature space while maximizing the distance between the closest points on either side of the hyperplane (Fig 5). This margin maximization produces a robust model, as demonstrated in linearly separable data examples.

In ideal scenarios, data are linearly separable, allowing for straightforward training of a linear SVM. However, real-world data often requires a more flexible approach. By introducing slack variables, SVMs can maximize the soft margin, permitting some classification errors and thereby improving the handling of nearly linearly separable data and reducing overfitting. For linearly inseparable data, SVMs utilize kernel functions to map data to higher-dimensional spaces, making them linearly separable.

Eleven of the included papers (n = 11, 52%) [20,31,34,38–43,46,48] utilized SVM for processing handwritten images, integrating additional features such as kinematic, temporal, and spatial data. The most effective kernels employed were the one-class SVM [42] and the Radial Basis Function (RBF) kernel [43]. Additionally, combining SVM with fusion models has been shown to be particularly effective in improving the performance of these systems [20]. In the context of dysgraphia diagnosis, SVMs excel in handling small, high-dimensional handwriting datasets, such as those incorporating kinematic, spatial, and temporal features from Wacom Tablets [20,43]. Their robustness to overfitting makes them particularly suitable for language-specific datasets with limited samples, as demonstrated by the use of one-class SVM and RBF kernels [42,43]. However, their high computational cost may pose challenges for real-time applications in educational or clinical settings, and performance relies on careful kernel selection to accommodate the variability in dysgraphic handwriting patterns. A summary of the advantages and disadvantages of the SVM algorithm for dysgraphia diagnosis is provided in Table 6.

**3.5.2. KNN.** The K-Nearest Neighbors (KNN) algorithm classifies data points by their closeness to other data points within the feature space [61]. For a given input, it identifies the K closest neighbors and assigns the most common

**Table 4. Overview of methodological approaches.**

| Study | Models | Feature Extraction | Fusion Model | Data Augmentation | Noise Removal | Normalization | Binarization | Dimensionality Reduction | Performance |
|---|---|---|---|---|---|---|---|---|---|
| [38] | **AdaBoost**/SVM/RF | WKNN_F5 | – | Zooming/ Shifting/ Shearing | – | Z-Score | – | – | ACC 79.7/ SPEC 76.7/ SEN 79.7 |
| [32] | **Directed Acyclic Graph** | Convolutional Layers | – | Rotation/ Resizing | – | Batch Normalization | – | Average Pooling | ACC 99/ PRE 98/ REC 99/ F1 99 |
| [41] | **CNN**/SVM/RF/NB/DT/ KNN | Convolutional Layers | – | Flipping | – | Min Max Scaling | – | Max Pooling | ACC 91/PRE 91/REC 91/ F1 95 |
| [39] | KNN/SVM/ RF/**AdaBoost** | Comprehensive Extraction | – | – | – | Min Max Scaling | – | Boost | ACC 80/ PRE 83/ REC 78/ F1 80/ ROC & AUC |
| [58] | CNN: RNN+LSTM | Convolutional Layers | – | Gaussian Blur/ Dilation/ Erosion | – | – | TRUE | – | ACC 73 |
| [20] | **SVM**/RF/AdaBoost | DenseNet201 | Horizontal Concatenation: All pairs & Triads | Zooming/ Shifting/ Shearing | – | – | – | – | ACC 97/ PREC 97/ REC 97/ F1 97/ AUC & ROC |
| [33] | CNN/**ResNet**/VGG16/ Inception V3 | Convolutional Layers | – | Mirroring | – | Resizing | – | Inception V3 | ACC 98/ SEN 95/ SPEC 93 |
| [42] | RF/SVM/LR/**One-Class SVM** | Contour Analysis/Average non-zero pixels | – | – | Bilateral Filtering | – | TRUE | – | ACC 92/ PREC 90/ REC 77/ F1 83 |
| [40] | **CNN**/AdaBoost/SVM/ RF | Interpolation/ DFT | – | – | – | Interpolation | – | – | ACC 79/TPR 76/ TNR 80 |
| [43] | Gaussian Process/**SVM**/LSTM/DT/ RF/NB/AdaBoost/Gradient Boost/ | Fisher Criterion/ SFFS | – | – | Butterworth Filter | Z-Score | – | – | ACC 86/SPEC 91/ SEN 81/ |
| [44] | **K-means (Binary)** | Divide into Bins/Signal Extraction/Fourier Transform | – | – | Butterworth Filter | Z-Score | – | PCA | SEN 91/ SPEC 90 |
| [34] | RF/**SVM** | TQWT | – | – | SNR | Vertical Normalized Jerks | – | mRMR | SEN 80/ SPEC 68/MCC 49 |
| [35] | RF/**SVM**/AdaBoost | Comprehensive Extraction | – | – | – | Z-Score | – | – | ACC 76/ SEN 75/ SPEC 76 |
| [36] | **RF** | GranWald-Letnikov Approximation | – | – | – | – | – | – | ACC 89/ SEN 88/ SPEC 90 |
| [45] | **RF** | Divide into Bins/ Fourier Transform | – | – | Gaussian Filter/ Fourier Transform/Moving Average Filter | Averaging Fourier Transforms | – | – | SEN 96/ SPEC 99/ F1 97 |
| [46] | **SVM** | Pen Displacement Analysis | – | – | Hann Kernel Filtering | Z-Score | – | – | ACC 89/ SPEC 90/ SEN 90/ ROC & AUC |

*(Continued)*

| Study | Models | Feature Extraction | Fusion Model | Data Augmentation | Noise Removal | Normalization | Binarization | Dimensionality Reduction | Performance |
|---|---|---|---|---|---|---|---|---|---|
| [37] | MobileNet/**CNN** | Bounding Boxes and Contours | – | – | Gaussian Blur | Contrast & Brightness Normalization | – | – | Classifying into Below Ave/ Ave/ Above Ave |
| [31] | **AdaBoost**/SVM/RF/KNN | Comprehensive Extraction | – | – | – | – | – | – | ACC 79 |
| [47] | **RF**/LDA | Comprehensive Extraction | Multidimensional Fusion Model/HPSQ | – | – | Cohort Normalization | – | mRMR/SFFS | ACC 96/ SPEC 96/ SEN 96/ TSS 1.99 |
| [48] | **NDR-R2CNN**/SVM/RF/DT | Convolutional Layers | – | Rotating/Scaling | Gaussian Blur | Z-Score | TRUE | – | ACC 98/ PREC 96/ REC 100/ F1 98 |
| [57] | **CNN** | Convolutional Layers | – | – | – | – | – | – | ACC 91/ PREC 93/ REC 91/ F1 91 |

\* Models in **BOLD** are the main model

\* Short Forms: CNN: Convolutional Neural Network/DT: Decision Tree/RF: Random Forest/LDA: Linear Discriminant Analysis/LR: Logistic Regression/AdaBoost: Adaptive Boosting/KNN: K Nearest Neighbors

**Table 5. Results of scoring.**

| Study | First Author | Year | Validity Score | Relevance Score | Result Score | Total Score | Category |
|---|---|---|---|---|---|---|---|
| [38] | Peter Drotar | 2020 | 2 | 3 | 1 | 6 | Moderate Quality |
| [32] | Siti Azura Ramlan | 2023 | 1 | 3 | 3 | 7 | High Quality |
| [41] | Richa Gupta | 2023 | 3 | 3 | 3 | 9 | High Quality |
| [39] | Jayakanth Kunhoth | 2023 | 2 | 3 | 2 | 7 | High Quality |
| [58] | Neha N. Doshi | 2023 | 1 | 3 | 1 | 5 | Moderate Quality |
| [20] | Jayakanth Kunhoth | 2023 | 2 | 3 | 3 | 8 | High Quality |
| [33] | Sharmila C | 2023 | 2 | 3 | 3 | 8 | High Quality |
| [42] | Basant Agarwal | 2023 | 3 | 3 | 3 | 9 | High Quality |
| [40] | Juraj Skunda | 2022 | 2 | 3 | 1 | 6 | Moderate Quality |
| [43] | Louise Deschamps | 2021 | 3 | 3 | 2 | 8 | High Quality |
| [44] | Thibault Asselborn | 2020 | 1 | 3 | 3 | 7 | High Quality |
| [34] | Vojtech Zvoncak | 2019 | 1 | 3 | 2 | 6 | Moderate Quality |
| [35] | Zuzana Dankovicova | 2019 | 2 | 3 | 1 | 6 | Moderate Quality |
| [36] | Jan Mucha | 2018 | 2 | 2 | 1 | 5 | Moderate Quality |
| [45] | Thibault Asselborn | 2018 | 1 | 3 | 3 | 7 | High Quality |
| [46] | Sara Rosenblum | 2017 | 1 | 3 | 2 | 6 | Moderate Quality |
| [37] | Tushar B. T. | 2024 | 1 | 2 | 0 | 3 | Low Quality |
| [31] | Jayakanth Kunhoth | 2022 | 2 | 3 | 1 | 6 | Moderate Quality |
| [47] | Jiri Mekyska | 2017 | 1 | 3 | 3 | 7 | High Quality |
| [48] | Fatima Ghouse | 2021 | 3 | 3 | 3 | 9 | High Quality |
| [57] | Soukaina Gouraguine | 2023 | 1 | 3 | 3 | 7 | High Quality |

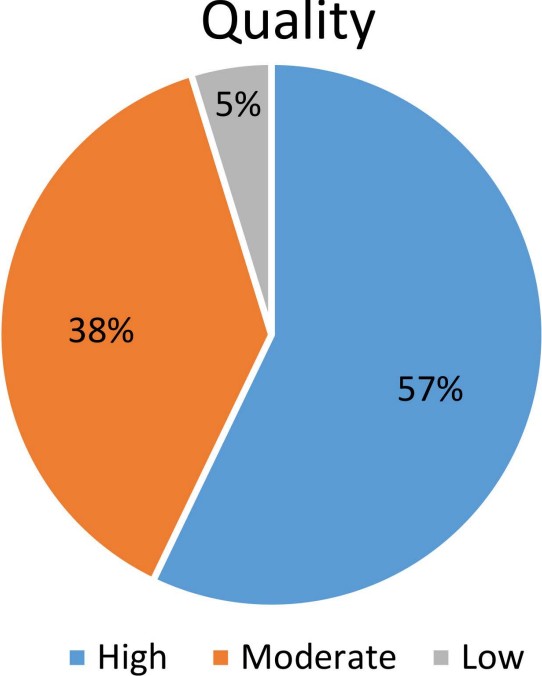

**Fig 4. Quality assessment scores of included studies.**

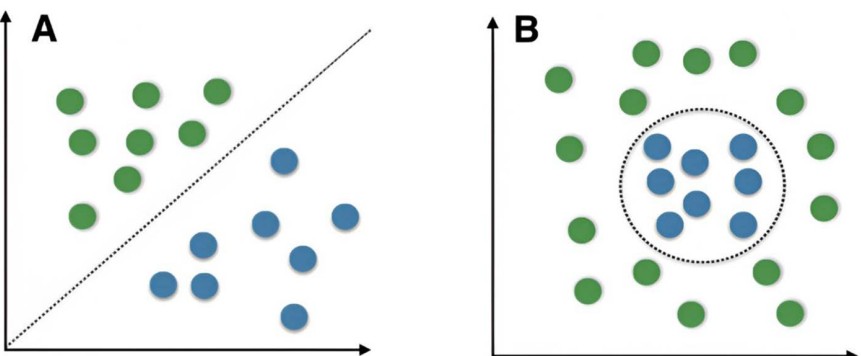

**Fig 5. Examples of linearly separable data distributions [60].**

**Table 6. Advantages and disadvantages of the SVM algorithm in dysgraphia diagnosis.**

| Advantages | Disadvantages |
|---|---|
| Effective for small, high-dimensional handwriting datasets | High computational cost and time consumption on large-scale datasets limit real-time use |
| Robust to overfitting in limited-sample, language-specific tasks | Kernel choice sensitivity affects performance on diverse handwriting |

label among them to the input. This method assumes data points with similar characteristics are located near each other, making it practical for various classification tasks, including image processing (Fig 6). KNN is non-parametric and straightforward to implement but can be computationally intensive with large datasets and sensitive to the choice of *K* and distance metrics.

In four of the included papers (n = 4, 19%) [38,39,31,41], KNN was utilized; in three of these, it served as a classifier, while in one, a variant called Weighted KNN_F5 [64] was used for feature extraction. Weighted KNN_F5 assigns different weights to neighbors based on their distance, giving closer neighbors more influence on the classification outcome. While the results of KNN as a classifier were not particularly strong, Weighted KNN F5 for feature extraction showed the potential to enhance the model's performance. For dysgraphia diagnosis, KNN's simplicity makes it a valuable baseline in early-stage studies, effectively classifying spatial and temporal handwriting features [38,39]. Variants like Weighted KNN F5 enhance feature extraction, improving performance with small datasets [64]. However, its sensitivity to noise, such as inconsistent stroke patterns in dysgraphic handwriting, limits its accuracy in complex cases [31]. Additionally, KNN's computational intensity with large or high-dimensional datasets poses challenges for scalable dysgraphia screening in educational settings. A summary of the advantages and disadvantages of the KNN algorithm for dysgraphia diagnosis is provided in Table 7.

**3.5.3. Random forest.** The Random Forest algorithm [65] is a versatile and robust ensemble learning technique for classification and regression tasks. It works by constructing multiple decision trees [66] during training and outputting the

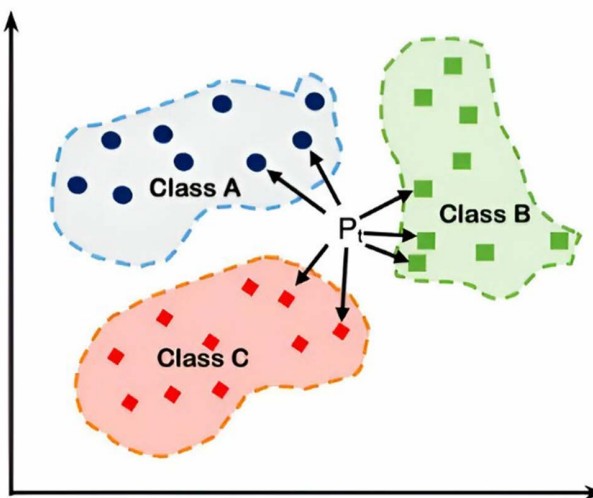

**Fig 6. KNN algorithm overview [62,63].**

**Table 7. Advantages and disadvantages of the KNN algorithm in dysgraphia diagnosis.**

| Advantages | Disadvantages |
|---|---|
| Simple baseline for early dysgraphia detection due to its simplicity and non-parametric nature | Sensitive to handwriting noise, reducing accuracy in complex dysgraphia tasks |
| Weighted variants improve performance on small datasets, benefiting from low training time | High cost on large/high-dimensional data due to memory usage, limiting real-world scalability |

mode of the classes (for classification) or mean prediction (for regression) of the individual trees (Fig 7). This approach leverages the wisdom of crowds, resulting in more accurate and robust predictions compared to a single decision tree.

Nine of the included papers (n = 9, 42%) [34–36,38,39,32,41,42,47] applied the Random Forest (RF) algorithm for dysgraphia detection based on images. While RF showed promising results when combined with other models, its performance was not as strong. In dysgraphia detection based on images, Random Forest is particularly effective because it can manage high-dimensional data and capture complex patterns [36,45]. In dysgraphia diagnosis, Random Forest's ability to handle mixed feature types, such as spatial, temporal, and pressure-based features, makes it highly effective for multimodal handwriting analysis, achieving robust performance in complex datasets, particularly valuable for diverse handwriting patterns in dysgraphia studies [36,45]. However, its computational intensity may hinder real-time screening in clinical or educational settings, and performance can vary with imbalanced datasets, necessitating careful feature selection [36]. A summary of the advantages and disadvantages of the Random Forest algorithm for dysgraphia diagnosis is provided in Table 8.

**3.5.4. K-means.** K-means is an unsupervised machine algorithm that groups data into clusters to identify distinct classes [68]. The algorithm involves four main steps: First, $K$ initial clustering centers are selected. Second, each sample in the dataset is allocated to the nearest cluster center based on distance calculations. Third, the cluster centers are

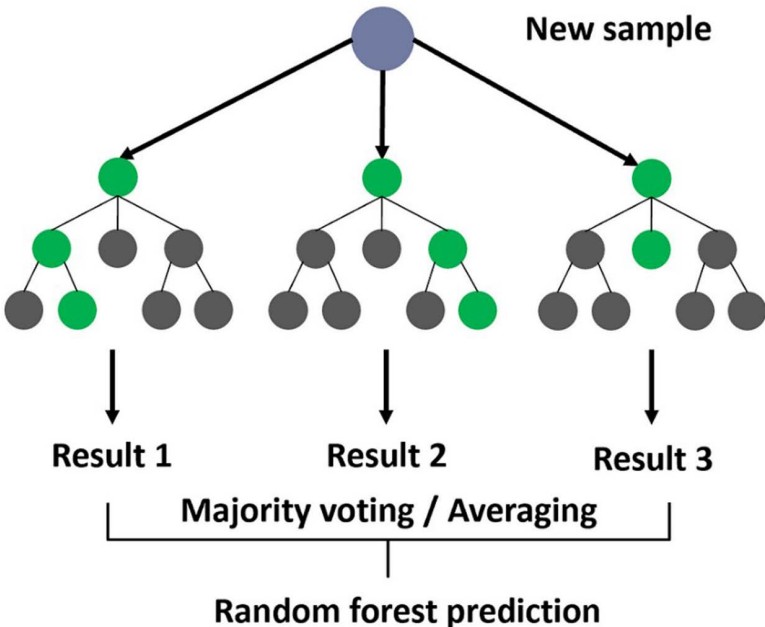

**Fig 7. Random forest process [67].**

**Table 8. Advantages and disadvantages of the random forest algorithm in dysgraphia diagnosis.**

| Advantages | Disadvantages |
|---|---|
| Handles mixed feature types and high-dimensional data well | Computationally intensive for large datasets; limits real-time screening |
| Boosts accuracy in multimodal handwriting analysis | Performance varies with imbalanced data; needs careful feature selection and may overfit without proper tuning |

recalculated for each group. Finally, the second and third steps are repeated until a specified stopping criterion, such as reaching the maximum number of iterations or achieving minimal changes in cluster centers, is met (Fig 8).

K-means can also be adapted for classification tasks by defining the number of clusters, *K*, to correspond to the number of classes [44]. After the clustering process, new data points can be classified by assigning them to the nearest cluster center. This approach leverages the simplicity and efficiency of K-means to create a basic classification model that groups data points based on their feature similarity to predefined cluster centers. In dysgraphia diagnosis, K-means is effective for clustering dysgraphic vs. non-dysgraphic handwriting patterns in exploratory studies, leveraging its simplicity for small datasets [44]. Its computational efficiency supports initial detection tasks. However, its performance is limited by small, imbalanced dysgraphia datasets, which may result in poor clustering of complex handwriting patterns [44]. Additionally, the need to predefine the number of clusters poses challenges for heterogeneous dysgraphia data with varied writing styles. A summary of the advantages and disadvantages of the K-means algorithm for dysgraphia diagnosis is provided in Table 9.

**3.5.5. AdaBoost.** AdaBoost, short for Adaptive Boosting, is a powerful ensemble learning algorithm for classification tasks, introduced by Yoav Freund and Robert Schapire in 1996 [70]. AdaBoost's core principle is to improve weak classifiers' accuracy by integrating them into a robust, strong classifier (Fig 9). It starts by assigning equal weights to all training samples and iteratively trains a sequence of weak classifiers, typically simple models like decision stumps. After each iteration, it evaluates the classifier's error rate and adjusts the sample weights, increasing the weights of

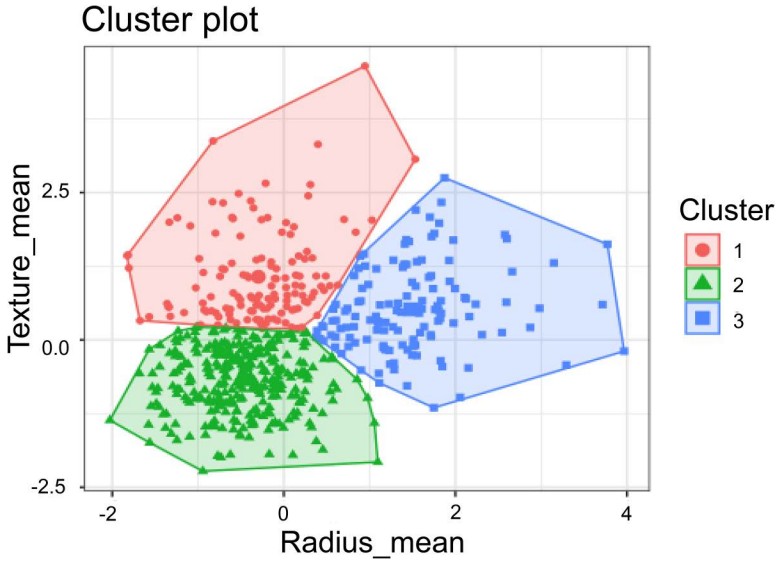

**Fig 8. The example of the K-means algorithm [69].**

**Table 9. Advantages and disadvantages of the K-means algorithm in dysgraphia diagnosis.**

| Advantages | Disadvantages |
|---|---|
| Effectively clusters dysgraphic vs. non-dysgraphic patterns | Struggles with imbalanced data and outliers; poor clustering of complex patterns |
| Simple, efficient, and easy to interpret for small handwriting datasets due to low complexity | Requires predefined cluster numbers; limited with heterogeneous or highly discrete classes |

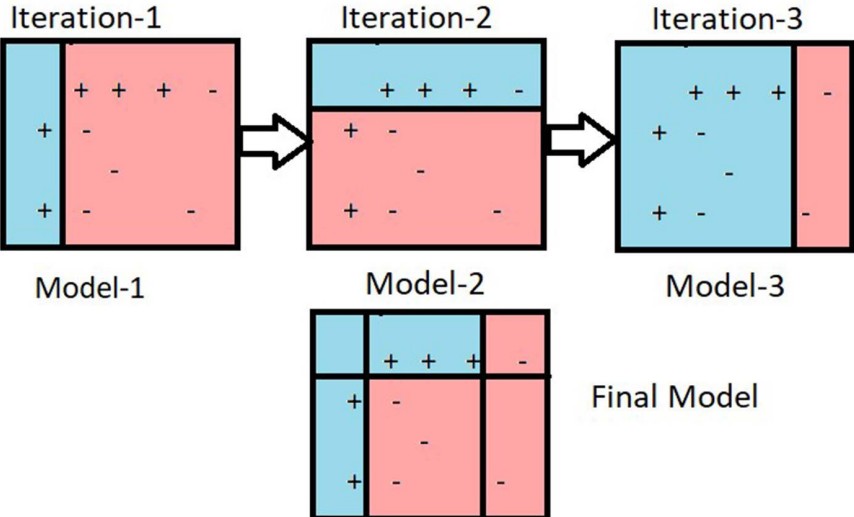

**Fig 9. The example of the AdaBoost process [71].**

misclassified samples to ensure subsequent classifiers focus on harder-to-classify cases. The final model is constructed by aggregating all the weak classifiers, where each classifier's influence is proportional to its accuracy. AdaBoost is known for its ability to significantly enhance model performance, its adaptability to various types of weak classifiers, and its robustness against overfitting, although it can be sensitive to noisy data and outliers. This adaptive focusing mechanism and iterative process make AdaBoost a powerful tool in machine learning for creating highly accurate classification models. Notably, seven of the included papers utilized AdaBoost (n = 7, 33%) [20,31,38–40,43,35], with three of them (n = 3, 14%) employing it as the primary model in their studies [38,39,47]. For dysgraphia diagnosis, AdaBoost enhances the accuracy of weak classifiers, making it effective for small handwriting datasets with diverse features like spatial and kinematic attributes [20,38,39]. Its robustness to overfitting supports its use in combining multiple handwriting features. However, its sensitivity to noisy data, such as irregular strokes in dysgraphic samples, may reduce performance in complex cases [20]. Additionally, the need for careful tuning of weak classifiers increases complexity for practical dysgraphia applications. A summary of the advantages and disadvantages of the AdaBoost algorithm for dysgraphia diagnosis is provided in Table 10.

**3.5.6. CNN.** Convolutional Neural Networks [72] are a type of deep learning algorithms [73] particularly well-suited for image processing tasks. CNNs are designed to automatically and adaptively learn hierarchical spatial features from input images. The architecture of a CNN typically consists of several layers, including convolutional layers, pooling layers, and fully connected layers (Fig 10). The convolutional layers perform convolution operations to the input image, using a set of learnable filters to produce feature maps that capture different input characteristics, such as edges, textures, and patterns. These feature maps are then passed through activation functions to introduce non-linearity, enhancing the model's ability

**Table 10. Advantages and disadvantages of the AdaBoost algorithm in dysgraphia diagnosis.**

| Advantages | Disadvantages |
|---|---|
| Boosts weak classifiers, improving accuracy in small, diverse handwriting datasets | Sensitive to noisy handwriting; lowers performance in complex cases |
| AdaptRobust to overfitting; adapts well to combined handwriting features | Requires careful tuning; adds computational complexity |

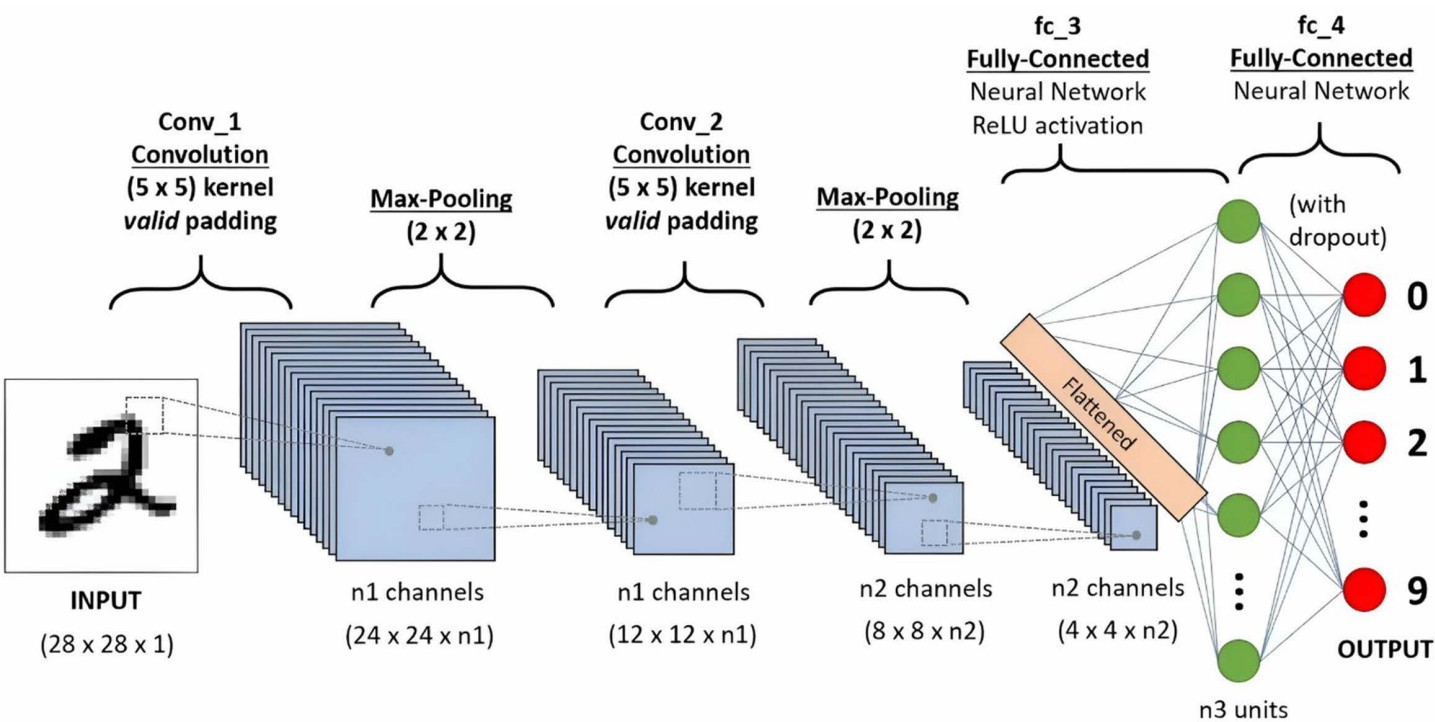

**Fig 10. Sample of a typical CNN [74].**

to learn complex representations. Pooling layers follow, performing down-sampling operations that shrink the spatial dimensions of the feature maps, thereby reducing computational requirements and providing some degree of translation invariance. Lastly, fully connected layers integrate the extracted features to perform the final classification or regression task. This hierarchical structure allows CNNs to learn more abstract and high-level features progressively, making them very effective for tasks such as image classification, object detection, and segmentation.

Seven of the included papers (n = 7, 33%) utilized CNNs, with two (n = 2, 9%) specifically employing well-known CNN-based models: one of the papers [37] using MobileNet [75] and the other paper [33] collectively incorporating Inception [76], VGGNet [77], and ResNet [78]. These studies achieved a high level of accuracy, with an average accuracy of 88% across all papers that used CNNs. Although CNNs are limited in capturing long-distance dependencies [52], they performed exceptionally well in most cases presented in the included papers. In dysgraphia diagnosis, CNNs excel in learning complex patterns from handwriting images, achieving high accuracy with models like MobileNet and ResNet [33,37]. Their ability to analyze spatial and kinematic features makes them ideal for robust image-based diagnosis. However, their reliance on large, diverse datasets is a challenge given the small sample sizes common in dysgraphia research [33]. Additionally, high computational demands may limit their use in resource-constrained educational or clinical settings. A summary of the advantages and disadvantages of CNNs for dysgraphia diagnosis is provided in Table 11.

Overall, the choice of an AI model for dysgraphia detection depends on dataset characteristics and application constraints. The trade-offs discussed highlight the importance of balancing model complexity, computational resources, and dataset limitations in achieving effective dysgraphia diagnosis.

**Table 11. Advantages and disadvantages of the CNN algorithm in dysgraphia diagnosis.**

| Advantages | Disadvantages |
|---|---|
| Computationally efficient with rapid training and high accuracy | Requires large, diverse handwriting datasets, often limited in dysgraphia research |
| Excels at learning complex handwriting patterns and spatial features | Less effective at capturing long-distance features, possibly missing global patterns |
| | High computational demands may limit use in resource-constrained settings |

## 3.6. Feature analysis

Feature selection is pivotal to the effectiveness of AI-based models for dysgraphia diagnosis. The quality, type, and modality of features extracted from handwriting samples significantly influence model accuracy and diagnostic reliability. This section analyzes features utilized in the reviewed studies, categorizing them by type, detailing their application in online and offline handwriting modalities, and evaluating their impact on model performance with specific examples.

### 3.6.1. Categorization of features.
AI models for dysgraphia leverage diverse features from handwriting samples, classified into four categories. Each category's role and application differ between online (real-time data, e.g., tablet-based writing) and offline (static data, e.g., scanned images) handwriting, reflecting distinct diagnostic insights.

**3.6.1.1. Temporal features:** Temporal features capture timing aspects of handwriting, such as stroke duration, inter-stroke pauses, and total writing time. In online handwriting, these features are directly measured via stylus or tablet devices, providing precise data on rhythm and pacing irregularities—hallmarks of dysgraphia. For example, temporal features can detect hesitations in letter formation, which are critical for identifying dysgraphic children who struggle with writing fluency. In offline handwriting, temporal features are inferred indirectly from stroke patterns or writing density, but their accuracy is limited due to the absence of real-time data. These features are particularly valuable in online settings for early detection, as they reveal subtle motor control issues not visible in static samples [38].

**3.6.1.2. Spatial features**: Spatial features encompass geometric characteristics, including letter size, shape, alignment, spacing between characters/words, and line adherence. In offline handwriting, spatial features are primary, extracted from scanned images to identify visual irregularities like inconsistent letter sizes or crowded spacing, common in dysgraphia. These features are robust for static analysis, as they rely on visible patterns rather than dynamic input. In online handwriting, spatial features are less dominant but still relevant, capturing real-time stroke geometry. Spatial features play a central role in both modalities for diagnosing dysgraphia by highlighting structural handwriting deficits, though offline applications are more prevalent due to their reliance on image-based data [79].

**3.6.1.3. Pressure-based features**: Pressure-based features measure the force exerted on the writing surface, including mean pressure, pressure variability, and peak force points. In online handwriting, these are captured directly by pressure-sensitive devices, revealing fine motor skill deficits (e.g., erratic pressure patterns indicating grip issues in dysgraphia). Pressure features are critical for assessing motor control precision, a key diagnostic marker. In offline handwriting, pressure features are rarely used, as scanned images lack force data, though some studies infer pressure from stroke thickness. These features are thus more significant in online modalities, enhancing models' ability to detect motor-related dysgraphia symptoms [31].

**3.6.1.4. Kinematic features**: Kinematic features quantify handwriting motion, including writing speed, acceleration, jerk (rate of acceleration change), and stroke smoothness. In online handwriting, kinematic features are richly captured via real-time tracking, enabling precise analysis of movement fluidity (e.g., jerky strokes in dysgraphic writers). These features are essential for detecting motor coordination issues. In offline handwriting, kinematic features are inferred from stroke trajectories or curvature, but their granularity is reduced due to static data limitations. Kinematic features are vital

in both modalities for assessing handwriting dynamics, with online applications offering superior detail for early diagnosis [39,31,79].

**3.6.2. Trends in feature selection.** The selection of features in AI models often depends on the study's specific objectives and the nature of the data available. Trends observed across the reviewed studies include:

a) **Online Handwriting:** Studies prioritize temporal and pressure-based features to capture real-time dynamics, such as irregular pauses or inconsistent pressure. For example, one study used temporal features to detect prolonged stroke durations, while another leveraged pressure variability to identify motor deficits [31].

b) **Offline Handwriting:** Spatial features dominate, focusing on letter shape, size, and spacing to detect visual irregularities. Kinematic features are less common but are used when stroke patterns allow motion inference [79].

c) **Hybrid Approaches:** Some studies integrate modalities, combining spatial features for offline analysis with temporal and kinematic features for online data, enhancing diagnostic accuracy. Hybrid models are increasingly common in studies aiming for comprehensive dysgraphia detection [21,38].

Additionally, the effectiveness of hybrid models depends on the consistency and precision of data acquisition tools. Variations in tablet sampling rates, resolution, or pressure sensitivity can introduce inconsistencies in feature definitions, impacting the reliability and generalizability of AI models across studies.

**3.6.3. Relationship between features and model performance.** Feature choice significantly impacts model performance, as demonstrated by reviewed studies:

a) Spatial and pressure-based features enhance accuracy in offline handwriting models. For instance, one study [79] achieved 91% accuracy using spatial features (letter size, spacing) in a CNN model, effectively capturing visual dysgraphia markers.

b) Temporal and kinematic features improve sensitivity in online handwriting models. One study [31] reported a 15% sensitivity increase in an SVM model by incorporating kinematic features (writing speed), aiding early motor difficulty detection.

c) Combining features across modalities boosts diagnostic efficacy. However, inconsistent feature definitions (e.g., varying pressure metrics across devices) hinder cross-study comparisons and standardization [21,38].

Furthermore, clinical interpretability of features remains a crucial aspect for real-world adoption. Features that align closely with known dysgraphia symptoms, such as prolonged stroke latency or erratic pressure spikes, can enhance clinician trust and provide actionable insights during assessment.

## 3.7. Strengths and challenges of studied papers

In this section, we examine the strengths and challenges of each reviewed paper, considering both the points explicitly mentioned by the authors and those we identified through our analysis of the papers (Table 12). This comprehensive evaluation clarifies each study's relative performance and effectiveness, allowing us to determine which approaches performed better on specific datasets. By highlighting both the advantages and limitations, we aim to offer a balanced view that facilitates more informed comparisons and insights into the efficacy of AI-based models in Dysgraphia detection through image processing tasks.

The primary challenges identified for state-of-the-art models include the high computational resources required, despite achieving accuracies exceeding 90% in all included papers (n = 5, 23%) utilizing modern techniques [20,32,45,47,48]. Another significant issue is the potential limitation in generalizability due to the small sample sizes. Moreover, language-specific models trained to predict particular languages pose a challenge, as their applicability may be limited to

**Table 12. Strengths and challenges of included papers.**

| Study | Strengths | Challenges |
|---|---|---|
| [38] | Comprehensive Dataset | Small Sample Size/Orthography Specific Language: Slovak |
| [32] | High Accuracy/Robustness/Great Sample Size | Variability in handwriting patterns among dysgraphia children |
| [41] | High Accuracy/Robust Data Augmentation Techniques | Potential variability or bias due to subjective classification criteria |
| [39] | Comprehensive Dataset and Extracted Features | Small Sample Size/Orthography Specific Language: Slovak/Combining On-Surface and In-Surface Features |
| [58] | High Accuracy/Advanced Techniques of Machine Learning | Small Sample Size |
| [20] | Comprehensive Dataset and Extracted Features | Small Sample Size/Orthography Specific Language: Slovak/Combining On-Surface and In-Surface Features |
| [33] | High Accuracy/ Use of Advanced Convolutional Models, such as VGG16, ResNet, and Inception | Small and Variable Dataset |
| [42] | Comprehensive Extracted Features/ High Accuracy | Difficulty in Data Collection |
| [40] | Comprehensive Dataset | Small Sample Size/Orthography Specific Language: Slovak/Combining On-Surface and In-Surface Features |
| [43] | Comprehensive Extracted Features | Small Sample Size/Tablet & Software Dependency/BHK Test Subjectivity |
| [44] | Low Computational Resource Required/Detecting the Severeness of Dysgraphia | Small & Heterogenous Dataset |
| [34] | Improve HD Identification Accuracy by Approximately 20% using TQWT Features, Effective Combination of Conventional and TQWT Features. | Small dataset/Subjectivity of Ratings in Dataset/Single Recording Session Limiting Intra-Writing Variability Analysis |
| [35] | Comprehensive Extracted Features/Visualization using PCA | Small Sample Size/Need for more Advanced Methods for Handwriting Processing and Evaluation. |
| [36] | Utilization of Fractional Derivatives in Kinematic Analysis, Achieving high Classification Accuracy with Minimal Features. | Small Number of Participants |
| [45] | Cost-Effectiveness/High Accuracy | Subjectivity in Assessment/Small Sample Size |
| [46] | High Accuracy/Applicability to Different Languages | Time-Consuming and Subjective Nature of Current Handwriting Evaluation Methods |
| [37] | Utilization of MobileNet for more Efficient Analysis/ Specific Focus on Kannada Handwriting | Offered Solutions are Time-Consuming/ Augmenting the Dataset with Specific Words Children Find Challenging Could Provide a More Targeted Training Dataset |
| [31] | Comprehensive Dataset/Multiple Combinations of Features Examined | Time Consuming/Small Sample Size/ Orthography Specific Language: Slovak |
| [47] | Variability in Tasks Done by Individuals | High Accuracy/Use of Simple Predictive Model (RF) |
| [48] | High Computational Resource Needed | High Accuracy/Comprehensive Extracted Features/Handled Imbalanced Data |
| [57] | Complexity of the Feature Extraction/Time-Costing and Subjectivity of the Data Collection Task | Great Sample Size/Innovative use of Interactive Robot/ |

those languages only. Subjectivity in data gathering and assessment processes further complicates the evaluation and comparability of results across studies.

Conversely, a notable strength of most included papers is the comprehensiveness of the extracted features, which encompass kinematic, spatial, temporal, and other relevant dimensions. This thorough feature extraction enhances the understanding and interpretation of the models' performance, contributing to more insightful and meaningful conclusions.

## 4. Discussion

Overall, we identified 21 studies on children under 18, especially school-going children, focusing on AI-based models for dysgraphia detection through handwritten image analysis, underscoring the need for early diagnosis and intervention. The review questions focused on identifying the most effective AI-based models, understanding the challenges associated with their application in dysgraphia detection, and suggesting future trends in dysgraphia detection models with higher performance. The evidence gathered highlights that AI models, particularly CNNs and SVMs, are the most commonly used and highly effective for this purpose, with accuracy rates often surpassing 90%. This aligns with the objectives of improving early diagnosis and intervention strategies for dysgraphia. These models effectively utilized both kinematic and spatial features extracted from handwriting samples, demonstrating high accuracy and robustness in classification tasks. Additionally, integrating fusion models and comprehensive feature extraction techniques further enhanced model performance.

A more nuanced interpretation of these results shows that while AI models such as CNNs and SVMs offer substantial accuracy, their effectiveness should be contextualized within current clinical practices. Clinicians must consider how these AI tools could mesh with existing diagnostic frameworks to enhance or streamline dysgraphia identification without supplanting necessary human judgment. This consideration is crucial for ensuring that AI complements rather than replaces traditional assessment methods, maintaining a balance between technological advancement and clinical intuition. Critically comparing different AI approaches, CNNs often excel in image-based analysis due to their ability to learn complex patterns from data, making them ideal for analyzing handwriting intricacies. However, SVMs might be preferred in scenarios with fewer data samples, or more straightforward decision boundaries are sufficient, highlighting a trade-off between model complexity and interpretability.

Across the studies, methodological limitations such as sample size, diversity, and validation methods have varied, impacting the generalizability of results. These variations suggest a need for standardized research protocols to better compare outcomes across studies. Moreover, potential biases in AI research, such as those introduced by the selection of datasets or the choice of performance metrics, can affect the perceived efficacy of these models. A critical examination of these biases is essential for advancing the field in a meaningful way. Finally, the gap between laboratory findings and real-world applications remains a significant challenge. The controlled environments in which many AI models are developed and tested do not always accurately reflect the complex realities of educational and clinical settings. Bridging this gap requires ongoing collaboration between developers, researchers, and practitioners to ensure these tools are adaptable and practical in everyday use.

These findings are particularly relevant to educational institutions, healthcare providers, and policymakers who manage learning disabilities. AI models can facilitate quicker and more accurate diagnoses for educators and healthcare providers, leading to timely and appropriate interventions. For policymakers, the evidence supports the potential for integrating AI-based diagnostic tools into educational and healthcare systems to help individuals with dysgraphia better.

### 4.1. Limitations

**4.1.1. Review process limitations.** Several limitations were identified during the scoping review process. The primary limitation was the availability of studies, with only 9 of the 21 included studies being open access. This restriction may have limited the comprehensiveness of the review, but fortunately, we accessed the other 12 studies by emailing the authors for access or purchasing the articles. Additionally, the sample sizes in many studies were small and often specific to certain languages, which may limit the generalizability of the findings.

**4.1.2. Extent of information uncovered.** The heterogeneity in evaluation metrics, such as accuracy, ROC, AUC, and MCC, across studies complicates direct comparisons of AI model performance in dysgraphia diagnosis. This variability underscores the urgent need for standardized metrics to facilitate consistent and comparable evaluations. Each metric serves distinct purposes in assessing model performance: Accuracy measures overall correctness but can be misleading in imbalanced datasets common in dysgraphia studies, where non-dysgraphic cases often outnumber dysgraphic ones. ROC curves illustrate a model's diagnostic ability across varying thresholds, AUC quantifies separability between classes, and MCC provides a balanced measure robust to class imbalance, making it particularly valuable for dysgraphia datasets with skewed distributions.

Trends in metric usage reveal a preference for accuracy and AUC in studies with large datasets, aiming to balance sensitivity and specificity, while MCC is increasingly adopted in imbalanced scenarios to ensure reliable detection of dysgraphic individuals. In clinical and educational contexts, these metrics have critical implications: high recall (sensitivity) is essential for school-based screening to minimize false negatives, which could delay interventions like occupational therapy for dysgraphic children, whereas high precision is prioritized in clinical settings to reduce false positives, avoiding unnecessary referrals and parental stress. For instance, a CNN-based screening tool deployed in educational platforms (e.g., via web-based systems) might prioritize recall to flag at-risk students, while a diagnostic tool for clinicians would emphasize precision to confirm cases.

These choices reflect meaningful trade-offs: false negatives risk delaying support for children in need, while false positives could lead to stress, mislabeling, and misallocated resources. Therefore, selecting metrics aligned with real-world diagnostic objectives is not merely technical—it directly impacts educational outcomes and patient well-being.

The implications of metric heterogeneity extend beyond academic challenges; they hinder the synthesis of findings and slow the development of standardized diagnostic models. To address this, future research should adopt a core metric set tailored to specific use cases, supported by public benchmark datasets to ensure fair and consistent comparisons. Best practices such as k-fold cross-validation, threshold tuning, and reporting multiple complementary metrics can further enhance metric reliability. Moreover, the development of interdisciplinary guidelines involving clinicians, educators, and AI researchers can help align evaluation criteria with practical diagnostic needs, ultimately fostering trust and real-world adoption of AI tools in dysgraphia diagnosis.

Another challenge lies in the incomplete or unclear information reported in some papers. For instance, certain studies omitted dataset details, such as size, requiring us to infer this from other papers using the same dataset. Additionally, reliance on self-reported data and the subjective nature of some assessments may introduce bias. Furthermore, the search was limited to major databases like PubMed, Scopus, Web of Science, IEEE Xplore, and SpringerLink, potentially missing relevant studies from other sources or unpublished works. Future studies could address this gap through broader, systematic searches to ensure a more comprehensive literature review.

**4.1.3. Methodological and research biases.** A more nuanced interpretation of the results in light of current clinical practices reveals that while AI models are highly effective under controlled study conditions, their application in typical clinical settings may not always align with traditional diagnostic methods. This misalignment necessitates a careful integration strategy to ensure that AI tools augment rather than replace the clinician's expertise. A critical comparison of different AI approaches, such as deep learning models like CNNs versus more traditional machine learning models like SVMs, highlights that while CNNs provide higher accuracy in image-based analyses due to their complex pattern recognition capabilities, SVMs offer advantages in terms of computational efficiency and transparency in decision-making. Examining methodological limitations across studies highlights issues such as the small and non-diverse sample sizes, which limit the extrapolation of findings to a broader population. Additionally, many studies do not undergo external validation, raising concerns about the robustness and reproducibility of the findings.

Discussion of potential biases in current research points to a prevalence of publication bias, where studies with positive findings are more likely to be published, and selection bias, where the data used may not adequately represent the

target population. An analysis of gaps between laboratory findings and real-world application indicates that AI models often perform exceptionally well in controlled environments but may fail to deliver similar results in real-world settings. This discrepancy can be attributed to the complex and variable nature of real-world environments compared to the controlled conditions of most studies.

## 4.2. Conclusions

This scoping review thoroughly examines the current landscape of AI-based models for dysgraphia detection through handwritten image analysis. The results indicate that AI models, particularly advanced machine learning models like CNNs and SVMs, can significantly enhance the diagnostic process for dysgraphia, offering more objective, faster, and more accurate assessments than traditional methods. This is crucial for mitigating the long-term impacts of dysgraphia on educational and social outcomes. This aligns well with the review objectives of identifying effective AI methodologies and understanding their application in real-world settings. Moreover, integrating machine learning techniques with traditional diagnostic methods has enhanced diagnostic accuracy and reliability synergistically. By incorporating machine learning outputs into conventional assessment frameworks, clinicians can better understand dysgraphia characteristics. The analysis of training datasets used in developing AI models revealed a significant variance in dataset quality and size, directly affecting model performance. Our review has identified a pressing need for standardized, large-scale datasets that more accurately reflect the diverse nature of handwriting styles across different demographics. Furthermore, ongoing technological advancements, such as the development of more sophisticated neural network architectures and enhanced computational power, promise to revolutionize the field of dysgraphia diagnosis further. These advancements will likely enable the creation of even more powerful and precise diagnostic tools in the near future.

## 4.3. Future recommendations

Before delving into future recommendations, it is essential to outline the primary roadmap for constructing a model for dysgraphia detection (Fig 11). The review also highlights the need for further research to address the limitations identified. Future studies should focus on issues such as more extensive and more diverse datasets, the development of language-independent models, and the standardization of evaluation metrics. The main steps to follow are listed below:

a) One of the most notable strategies is the combination of individual fusion models with traditional machine learning models. This approach has shown remarkable potential, as evidenced by two of the included studies, which achieved impressive accuracy rates of 97% [20] and 96% [47], respectively. This synergy between fusion and traditional models leverages the strengths of both methods, enhancing overall performance and robustness. Integration is typically achieved by extracting handcrafted or statistical features through traditional preprocessing pipelines, such as kinematic analysis or pressure curve interpretation, and feeding them alongside fused deep features into conventional classifiers like SVMs or Random Forests. While this hybrid design can capitalize on domain-specific insights and model generalizability, it may face limitations in scalability, increased computational complexity, and the challenge of optimally weighting heterogeneous feature sources [5,21,80,81]. In addition to developing new models, there is a critical need to enhance the robustness of machine learning integration with traditional diagnostic tools. This can be achieved by refining the algorithms that process the input from these tools, improving the synergy between AI models and existing diagnostic methods.

b) Another critical area for future work is addressing the limitation of inadequate datasets, which significantly impacts the training and evaluation of AI-based dysgraphia detection models. Many included studies relied on small or improperly structured datasets, often limited to specific languages, hindering model generalizability. To overcome this, advanced data augmentation methods can be used to enrich the datasets and enhance model performance substantially. In this review, seven papers utilized data augmentation techniques, achieving an average accuracy of 91%. The most

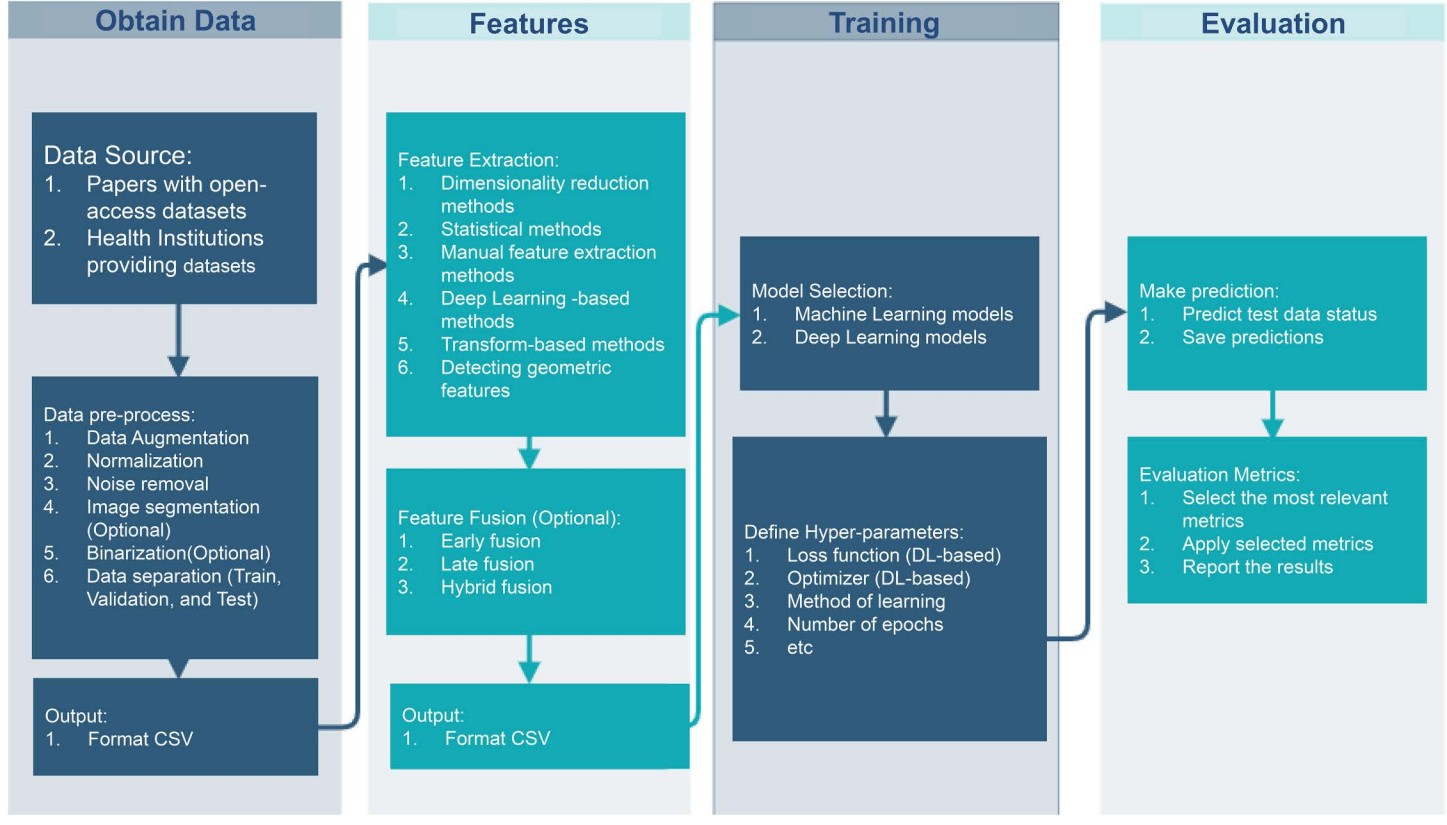

**Fig 11. Roadmap of dysgraphia detective model construction.**

effective augmentation methods identified include shearing, shifting, mirroring, and flipping. These techniques are particularly well-suited for handwriting data in the context of dysgraphia, as they simulate natural variations in writing style, stroke pressure, and spatial alignment—all of which are critical indicators in dysgraphia detection [20,21].

Equally important is the need to enhance dataset diversity across demographics, languages, and cultural contexts to ensure models are robust and equitable. Future studies should prioritize multi-center collaborations involving educational and healthcare institutions worldwide to collect handwriting samples from diverse populations, including varied age groups, genders, socioeconomic backgrounds, and educational levels. Including non-Latin writing systems, such as Arabic, Devanagari, or Chinese, is crucial, as handwriting characteristics vary significantly across scripts [9,44]. For example, partnerships with organizations like the Association of Dyslexia Malaysia could provide access to region-specific samples, while open-access repositories, such as the IAM Handwriting Database, can promote data sharing and inclusivity [45]. Cultural influences, such as writing conventions shaped by regional educational practices, should also be considered to capture context-specific handwriting variations [46,47]. These efforts will reduce biases inherent in language-specific or demographically limited datasets, improving model applicability in diverse clinical and educational settings globally [21,31]. By combining diverse datasets with augmentation techniques, future models can achieve higher generalizability, supporting equitable dysgraphia diagnosis and aligning with the need for standardized, inclusive diagnostic tools.

c) Additionally, the application of CNNs has outperformed other models, particularly when deeper CNNs are used. These deeper networks benefit from balancing parameters and incorporating non-discrimination regularization, which enhances

their learning and generalization capabilities. However, it is worth noting that such architectures often come with increased computational complexity and higher data requirements, which may limit their feasibility in low-resource or real-time diagnostic environments. The superior performance of CNNs, especially in handling complex datasets, underscores their importance in future research efforts, provided that considerations around efficiency and scalability are also addressed.

Overall, future research should focus on integrating fusion models with traditional ML approaches, employing advanced data augmentation techniques, prioritizing dataset diversity to ensure inclusivity across demographics, languages, and cultures, and leveraging the strengths of deeper CNN architectures. These steps will likely address current limitations and pave the way for more accurate and robust models.

### 4.4. Potential implications

These findings have substantial implications for both the educational and healthcare sectors. Integrating AI-based diagnostic tools could revolutionize how dysgraphia is identified and managed, leading to better academic outcomes and support for affected individuals. Furthermore, using these tools can help reduce the stigma associated with dysgraphia by providing clear, objective evidence of the condition.

In conclusion, while AI-based models promise to improve dysgraphia detection, ongoing research and development are essential to enhance their accuracy, generalizability, and practical application in real-world educational and healthcare settings.

## Supporting information

**S1 Table. This supplementary Excel file contains comprehensive data supporting the systematic review and meta-analysis described in the manuscript.**
(XLSX)

**S1 File. Supplementary data.**
(DOCX)

## Author contributions

**Conceptualization:** Hedieh Sajedi.

**Investigation:** Hedieh Sajedi.

**Methodology:** Avisa Fallah, Yazdan ZandiyeVakili, Hedieh Sajedi.

**Supervision:** Hedieh Sajedi.

**Validation:** Avisa Fallah, Yazdan ZandiyeVakili, Hedieh Sajedi.

**Visualization:** Avisa Fallah, Yazdan ZandiyeVakili.

**Writing – original draft:** Avisa Fallah, Yazdan ZandiyeVakili.

**Writing – review & editing:** Hedieh Sajedi.

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
