## [Decision Letter · Decision Letter 0]

23 Sep 2024

Dear Dr. Sajedi,

Thank you for submitting your manuscript to PLOS ONE. After careful consideration, we feel that it has merit but does not fully meet PLOS ONE’s publication criteria as it currently stands. Therefore, we invite you to submit a revised version of the manuscript that addresses the points raised during the review process.

We look forward to receiving your revised manuscript.

Kind regards,

Sadiq H. Abdulhussain, Ph.D.

Academic Editor

PLOS ONE

3. We note that Figure 11 in your submission contain copyrighted images. All PLOS content is published under the Creative Commons Attribution License (CC BY 4.0), which means that the manuscript, images, and Supporting Information files will be freely available online, and any third party is permitted to access, download, copy, distribute, and use these materials in any way, even commercially, with proper attribution. For more information, see our copyright guidelines: http://journals.plos.org/plosone/s/licenses-and-copyright.

1. You may seek permission from the original copyright holder of Figure 11 to publish the content specifically under the CC BY 4.0 license.

5. As required by our policy on Data Availability, please ensure your manuscript or supplementary information includes the following:

Reviewers' comments:

Reviewer's Responses to Questions

**Comments to the Author**

1. Is the manuscript technically sound, and do the data support the conclusions?

Reviewer #1: Partly

Reviewer #2: Partly

2. Has the statistical analysis been performed appropriately and rigorously?

Reviewer #1: Yes

Reviewer #2: N/A

3. Have the authors made all data underlying the findings in their manuscript fully available?

Reviewer #1: Yes

Reviewer #2: Yes

4. Is the manuscript presented in an intelligible fashion and written in standard English?

Reviewer #1: No

Reviewer #2: Yes

Reviewer #1: The Title: AI-Based Models for Dysgraphia Diagnosis on Handwritten Images: A Comprehensive Scoping Review.

This paper provides a scoping review to explore available studies employing AI-based models for dysgraphia diagnosis and identify the highest performance models and their challenges to suggest future improvements. The work is good and interesting in its field. However, there are some points need to be considered which are as follows:

1- The authors stated that a scoping review is conducted instead of a systematic review with three mentioned objectives. However, scoping review needs to identify knowledge gaps, explain concepts and investigate research conduct. Are these concepts addressed in this manuscript? The authors should elaborate further to show what has been added to the academic field and what has been discovered and proven.

2- It stated that “All original research articles published until April 2024 …….. were included.”.

Is there only one research paper in 2024 submitted in this manuscript? Please review this issue.

3- Some research questions need to be added to the manuscript. This type of review is broad because its purpose is to identify the scope of the literature on the presented topic, so, the research questions that this review can answer should be broad also.

4- Some references should be formatted according to the author's guidelines as there are many references that are not written correctly.

5- There is no need to add [30] in the title of Figure 2 because the information provided is the author’s own. Instead, [30] can be cited in the text of Section 3.1.

6- Figure 3 with the captions of its sub-figures needs more organization. In fact, the titles of most of the tables and figures needs to be reworded to make it more clear to the reader.

7- Ratios listed in figures that containing a circle should be as a percentage like the one in Figure 4.

8- There are many grammatical errors that should be checked and corrected in the entire manuscript.

9- Figure 5 can be presented in another representation and the resolution of Figure 6 and Figure 7 should be increased.

10- The section of the refences should have a title.

Reviewer #2: This paper presents a comprehensive scoping review about AI based approaches in literature proposed for Dysgraphia diagnosis from handwritten data. The paper needs major improvements before publishing. Here are my recommendations:

1. The current title, "AI-Based Models for Dysgraphia Diagnosis on Handwritten Images: A Comprehensive Scoping Review," may not accurately reflect the full scope of the paper. While the review covers both offline (image-based) and online handwriting data methods, the title only mentions "handwritten images," which typically refers to offline data. Consider revising the title to encompass both types of data.

For example: "AI-Based Models for Dysgraphia Diagnosis Using Online and Offline Handwriting Analysis: A Comprehensive Scoping Review"

2. While your scoping review provides insights into AI-based models for dysgraphia diagnosis, it currently lacks a discussion of existing review/survey papers on this topic. This omission limits the reader's ability to understand how your work builds upon or differs from previous efforts in the field. Include a section that identifies and briefly summarizes existing review papers related to AI-based dysgraphia diagnosis.

3. There are several instances in the paper where specific studies or data points are mentioned without proper citation. For example:

"Two of the papers utilized multiple datasets (n = 2, 9%), while only three papers used raw handwritten images (n = 3, 14%)"

This statement provides specific numbers and percentages but fails to cite the actual papers being referenced. This oversight occurs in other parts of the manuscript as well. Carefully review the entire manuscript to identify and address all similar instances where specific studies or data points are mentioned without proper citation.

4. What is the need of figure 5. If you want to include it, then please redraw and represent it in a better way.

5. While the acknowledgment of metric heterogeneity and the table of performance indicators are valuable, this crucial aspect warrants deeper analysis. Consider expanding this section to include: brief explanations of each metric and their relevance to dysgraphia diagnosis; trends in metric usage across studies; implications of this heterogeneity for the field's progress; context-specific considerations for dysgraphia evaluation; and potential future directions for standardization. Additionally, discuss how metric choices relate to clinical relevance and practical application. Enhancing this discussion will provide readers with a more comprehensive understanding of the challenges and opportunities in evaluating AI methods for dysgraphia diagnosis, significantly strengthening your scoping review.

6. The review provides a comprehensive overview of AI algorithms used in dysgraphia detection. However, a critical component is missing: an in-depth analysis of the features used in these models. Given that many methods employ traditional machine learning approaches, as mentioned in your review, a thorough examination of input features is essential. This analysis would significantly enhance the review's value and align with your stated goals of identifying effective AI-based models and understanding associated challenges. Add a dedicated section on feature analysis. Categorize and discuss types of features used (e.g., temporal, spatial, pressure-based, kinematic). Analyse trends in feature selection across studies. Examine the relationship between feature choices and model performance.

7. While the current review provides valuable insights into AI-based models for dysgraphia diagnosis, the Discussion and Limitations sections could benefit from deeper analysis and more comprehensive treatment. These sections are crucial for contextualizing findings, critically evaluating the current state of research, and guiding future work. Consider expanding these sections to include: a more nuanced interpretation of the results in light of current clinical practices; a critical comparison of different AI approaches and their relative strengths and weaknesses; a thorough examination of methodological limitations across studies; discussion of potential biases in current research; and analysis of gaps between laboratory findings and real-world application.

Minor comments :

Improve the presentation language and make sure the manuscript is free from typos

**Do you want your identity to be public for this peer review?** For information about this choice, including consent withdrawal, please see our Privacy Policy

Reviewer #1: No

Reviewer #2: No

---

## [Author Response · Author response to Decision Letter 1]

1 Mar 2025

Note: To enhance clarity, we have highlighted the reviewers' comments and used Arial font for them, while our responses are provided in Times New Roman.

We have carefully reviewed the PLOS ONE formatting guidelines and ensured that our manuscript adheres to the required style, including file naming conventions.

The datasets are public. The links for download are available in the data availability section.

3. We note that Figure 11 in your submission contain copyrighted images. All PLOS content is published under the Creative Commons Attribution License (CC BY 4.0), which means that the manuscript, images, and Supporting Information files will be freely available online, and any third party is permitted to access, download, copy, distribute, and use these materials in any way, even commercially, with proper attribution. For more information, see our copyright guidelines: http://journals.plos.org/plosone/s/licenses-and-copyright.

1. You may seek permission from the original copyright holder of Figure 11 to publish the content specifically under the CC BY 4.0 license.

“I request permission for the open-access journal PLOS ONE to publish XXX under the Creative Commons Attribution License (CCAL) CC BY 4.0 (http://creativecommons.org/licenses/by/4.0). Please be aware that this license allows unrestricted use and distribution, even commercially, by third parties. Please reply and provide explicit written permission to publish XXX under a CC BY license and complete the attached form.”

We have addressed this issue by removing and replacing the copyrighted image with an open-access image from this source. The new figure adheres to the CC BY 4.0 license requirements. We have also updated the figure caption accordingly.

Before:

Fig 11 Sample of a typical CNN [67]

After:

Fig 10 Sample of a typical CNN [67]

We have addressed this by including the captions for all Supporting Information files in a separate file named "DysgraphiaPaper_Files", following the journal’s guidelines.

5. As required by our policy on Data Availability, please ensure your manuscript or supplementary information includes the following:

We have addressed this by including all required tables and explanations in our supplementary file titled 'Supplementary-Data.' This file contains comprehensive data supporting our systematic review and meta-analysis, organized into nine sheets covering literature search, study selection, and data extraction. It includes a complete list of screened studies, reasons for inclusion/exclusion, keyword lists, search strategies, database-specific extractions, and a detailed breakdown of data extraction responsibilities. Additionally, we provide explanations on quality assessment, data privacy compliance, and accessibility instructions. All figures from the manuscript are included separately in the supplementary files, with proper references provided.

For clarity, the detailed contents of 'Supplementary-Data' are also provided below:

Supplementary Data Explanation

Excel file

This supplementary Excel file contains comprehensive data supporting the systematic review and meta-analysis described in the manuscript. The file is organized into nine sheets, each providing specific information related to the literature search, study selection, and data extraction processes employed in our study. Here is a detailed overview of each sheet:

Sheet 'Paper': This sheet contains a consolidated list of all studies that were initially screened during the literature search. It includes important information such as study IDs, authors, titles, publication years, and outcomes. This comprehensive listing helps in understanding the scope of literature considered for the review.

Sheet 'Included_Paper': This sheet details the studies that were included in the final analysis. It features a color-coded system where entries marked in blue represent data extracted by Yazdan Zandiye Vakili, and entries in white are extracted by Avisa Fallah. This differentiation aids in recognizing the contributions of individual team members to the data extraction process.

Sheet 'Keyword': It presents the finalized list of keywords that were used to perform the literature search across various databases. This sheet is crucial for replicating the search methodology used in our study.

Sheet 'Source': Contains detailed information about the selected sources for the literature search, including direct links to the search queries applied and specific search strategies used for each source.

Sheets 'IEEE_Xplore', 'PubMed', 'Springer', 'Scopus', 'WOS' (World of Science): Each of these sheets provides specific details on the papers extracted from the respective databases. The sheet for each database outlines which papers were considered and includes notes on the extractor, further illustrating the distribution of work among team members.

Additional Explanations:

Access Instructions: This supplementary material is available directly via contacting the corresponding author.

Inclusion and Exclusion Reasons: All the papers, as shown in the PRISMA figure, are categorized based on whether they are review papers, related to the topic, or ultimately included or excluded. Some of the excluded papers were not specifically about dysgraphia itself. Instead, they focused on related topics, such as analyses involving EEGs or brain images, rather than handwritten images. Others were excluded because they were review papers rather than empirical studies.

Quality assessment: The quality of the included studies is assessed in the paper and all the explanations can be found in the section 3.4 (Results of Critical Appraisal within Sources of Evidence).

Data Protection and Privacy Compliance: All data included in this file have been anonymized and prepared in compliance with applicable data protection and privacy regulations. No personal or sensitive information is disclosed.

Version and Date: This document is the final version, created on 20/03/2024, and the final updates were applied on 27/05/2024. For any questions or corrections post-publication, please contact the corresponding author.

Use and Navigation Tips: Users are encouraged to review the 'Sheet Guide' included within the file for tips on navigating and understanding the contents effectively. Each sheet is designed to be self-explanatory, with headings and notes providing guidance on the data presented.

Figures

All the figures in the paper, are included separately in the supplementary files. All of the eleven figures are either created by us or is open accessed and you can see the reference of that in the paper.

Reviewers' comments:

Reviewer's Responses to Questions

Comments to the Author

4. Is the manuscript presented in an intelligible fashion and written in standard English?

Reviewer #1: No

Reviewer #2: Yes

We have revised the manuscript to improve clarity, correctness, and readability. This includes refining the text, enhancing image explanations, and correcting grammatical and scientific errors. The manuscript is now presented in a clear, correct, and unambiguous manner.

5. Review Comments to the Author

Reviewer #1: The Title: AI-Based Models for Dysgraphia Diagnosis on Handwritten Images: A Comprehensive Scoping Review.

This paper provides a scoping review to explore available studies employing AI-based models for dysgraphia diagnosis and identify the highest performance models and their challenges to suggest future improvements. The work is good and interesting in its field. However, there are some points need to be considered which are as follows:

1- The authors stated that a scoping review is conducted instead of a systematic review with three mentioned objectives. However, scoping review needs to identify knowledge gaps, explain concepts and investigate research conduct. Are these concepts addressed in this manuscript? The authors should elaborate further to show what has been added to the academic field and what has been discovered and proven.

& 

3- Some research questions need to be added to the manuscript. This type of review is broad because its purpose is to identify the scope of the literature on the presented topic, so, the research questions that this review can answer should be broad also.

We have made several important revisions to ensure the manuscript addresses these concerns effectively:

Clarification on Scoping Review Objectives:

We have added three new objectives (4, 5, and 6) to the introduction to further elaborate on the contributions of this review to the academic field and highlight the gaps discovered:

Objective 4: Examine the integration of machine learning techniques with traditional diagnostic methods.

This objective investigates how machine learning techniques are integrated with traditional diagnostic methods for dysgraphia, identifying the benefits and drawbacks of such integrations. It includes an analysis of hybrid models that combine AI with manual assessments or other diagnostic tools.

Objective 5: Analyze the training datasets used for developing AI models in dysgraphia diagnosis.

This objective analyzes the characteristics, diversity, and adequacy of the training datasets used in AI model development for dysgraphia diagnosis. It addresses the quality of data and its impact on model performance and generalizability.

Objective 6: Review technological advancements and their implications for future AI models in dysgraphia diagnosis.

This objective reviews recent technological advancements in AI and machine learning, including algorithm advancements, computational hardware, and data collection methods, and discusses their potential impacts on future models for dysgraphia diagnosis.

Before:

Objectives:

1- Identify available studies that used AI-based models for dysgraphia diagnosis, explore the current dominant use of AI in expediting dysgraphia detection.

2- Explore the challenges of building computational AI-based models under current conditions and the limitations of each study.

3- Determine the best models with high accuracy and performance in dysgraphia diagnosis and examine the effectiveness of these models in dysgraphia detection and possible future improvements in these models.

After:

Objectives:

1- Identify available studies that used AI-based models for dysgraphia diagnosis and explore the current dominant use of AI in expediting dysgraphia detection.

2- Explore the challenges of building computational AI-based models under current conditions

---

## [Decision Letter · Decision Letter 1]

7 Apr 2025

Dear Dr. Sajedi,

Thank you for submitting your manuscript to PLOS ONE. After careful consideration, we feel that it has merit but does not fully meet PLOS ONE’s publication criteria as it currently stands. Therefore, we invite you to submit a revised version of the manuscript that addresses the points raised during the review process.

We look forward to receiving your revised manuscript.

Kind regards,

Sadiq H. Abdulhussain, Ph.D.

Academic Editor

PLOS ONE

Reviewers' comments:

Reviewer's Responses to Questions

**Comments to the Author**

Reviewer #1: All comments have been addressed

Reviewer #2: (No Response)

2. Is the manuscript technically sound, and do the data support the conclusions?

Reviewer #1: Yes

Reviewer #2: Partly

3. Has the statistical analysis been performed appropriately and rigorously?

Reviewer #1: I Don't Know

Reviewer #2: N/A

4. Have the authors made all data underlying the findings in their manuscript fully available?

Reviewer #1: Yes

Reviewer #2: Yes

5. Is the manuscript presented in an intelligible fashion and written in standard English?

Reviewer #1: Yes

Reviewer #2: Yes

Reviewer #1: Most of the comments in the revised version have been properly addressed and no further comments are needed.

Reviewer #2: The authors have made commendable efforts to revise the manuscript in response to my previous comments, and several aspects of the paper have indeed improved. However, I believe further refinements are necessary. Some of the earlier comments appear to have been addressed only in a general or superficial manner, lacking the specificity and depth required to significantly enhance the quality. Therefore, additional revision is recommended. In this round of feedback, I have provided more detailed and structured suggestions to help strengthen the manuscript further. Additionally, I have included one new comment (Comment number 5).

1. Regarding your response to my second comment in the first revision:

My comment was : "While your scoping review provides insights into AI-based models for dysgraphia diagnosis, it currently lacks a discussion of existing review/survey papers on this topic. This omission limits the reader's ability to understand how your work builds upon or differs from previous efforts in the field. Include a section that identifies and briefly summarizes existing review papers related to AI-based dysgraphia diagnosis."

While I appreciate the addition of a discussion on previous review papers in Section 3.2, the response remains too general and does not fully address the concern. The discussion broadly mentions limitations of past reviews without explicitly citing and engaging with specific existing surveys on AI-based dysgraphia diagnosis. For example, one recent and comprehensive review (https://link.springer.com/article/10.1007/s10032-024-00464-z) has examined AI methodologies for dysgraphia diagnosis, and provided a structured analysis of automated systems, yet your manuscript does not acknowledge or differentiate itself from this prior work. Instead, it makes broad claims about the shortcomings of earlier reviews without demonstrating how your study adds new insights beyond what has already been explored. To strengthen the scholarly rigor of your manuscript, I recommend explicitly citing and summarizing existing review papers rather than referring to them in a generalized manner, clearly articulating how your review extends, updates, or differs from prior work, and avoiding generalized critiques of previous reviews unless supported by direct comparisons with specific papers. A more detailed engagement with prior surveys will provide a clearer justification for your review and ensure it is properly positioned within the existing body of literature.

Also i suggest changing the title of section 3.2.

2. While the added discussion on evaluation metrics (Section 4.1.2) is a valuable improvement, the section would benefit from further expansion on the clinical and practical implications of metric choices. Specifically, it would be useful to discuss how different metrics align with real-world dysgraphia diagnosis scenarios, considering the trade-offs between false positives and false negatives. Additionally, while the discussion acknowledges metric heterogeneity, it would be strengthened by proposing potential future directions for standardization and best practices in evaluating AI models for dysgraphia.

3. While the inclusion of a dedicated section (3.6) on feature analysis is a valuable addition, I believe there is still an opportunity to provide more depth and specificity. The categorization of features into temporal, spatial, pressure-based, and kinematic is a useful framework, but a more detailed discussion of how these features vary in the context of both online and offline handwriting could add significant value. Given the differences between these two modalities, it would be helpful to explore trends in feature selection across studies focusing on each type of handwriting separately. Additionally, discussing the relationship between feature selection and model performance in more detail—perhaps with specific examples from the reviewed studies—would help clarify how different features contribute to model efficacy.

Including relevant citations to support the claims made in this section would also strengthen the discussion, grounding the analysis in the broader body of literature. A brief overview of challenges related to feature standardization across studies would further enrich this section and highlight potential directions for future research. Overall, while the section is a good starting point, expanding on these aspects will provide readers with a more comprehensive understanding of the role that feature selection plays in the development of AI-based dysgraphia diagnostic models.

4.The recommendations outlined in future works section provide a solid foundation for advancing dysgraphia detection models. The focus on combining individual fusion models with traditional machine learning approaches, addressing dataset limitations, and leveraging CNNs for improved performance are highly relevant and aligned with current research trends. However, a few aspects could benefit from further elaboration. For example, while the synergy between fusion models and traditional methods is well-highlighted, additional context or examples on how such integration could be practically implemented would strengthen this point. It would also be useful to address potential challenges, such as computational complexity or data requirements, associated with this integration. Regarding dataset limitations, the mention of data augmentation methods is insightful, but a more detailed explanation of how these techniques can be specifically applied to handwriting data in the context of dysgraphia would be beneficial. Additionally, discussing how future studies could ensure that datasets are more diverse in terms of demographics, language, and cultural context would help provide a broader perspective.

Moreover, referencing studies that have successfully employed fusion models, advanced data augmentation, or CNNs in handwriting or related tasks could give a clearer picture of how these methods can be translated into dysgraphia detection. This would also demonstrate that the proposed future work is well-supported by the current body of research, giving it more credibility and a stronger basis for consideration.

5. While Tables 6 to 11 present general advantages and disadvantages of the SVM, KNN, Random Forest algorithms etc..., they would benefit from a more focused discussion contextualized to dysgraphia diagnosis. For example, SVM’s effectiveness with high-dimensional but small handwriting datasets has been reported in [Ref], while KNN’s simplicity makes it a common baseline in early-stage studies despite its sensitivity to noise. Random Forest’s ability to handle mixed feature types (e.g., spatial, temporal, pressure-based) has proven useful in multimodal handwriting analysis [Ref]. Including such application-specific insights would enhance the relevance of these comparisons and help readers understand the trade-offs in algorithm selection for this domain.

**Do you want your identity to be public for this peer review?** For information about this choice, including consent withdrawal, please see our Privacy Policy

Reviewer #1: No

Reviewer #2: No

---

## [Author Response · Author response to Decision Letter 2]

21 May 2025

Third Revision

Reviewer #1: Most of the comments in the revised version have been properly addressed and no further comments are needed.

Reviewer #2: The authors have made commendable efforts to revise the manuscript in response to my previous comments, and several aspects of the paper have indeed improved. However, I believe further refinements are necessary. Some of the earlier comments appear to have been addressed only in a general or superficial manner, lacking the specificity and depth required to significantly enhance the quality. Therefore, additional revision is recommended. In this round of feedback, I have provided more detailed and structured suggestions to help strengthen the manuscript further. Additionally, I have included one new comment (Comment number 5).

1. Regarding your response to my second comment in the first revision:

My comment was : "While your scoping review provides insights into AI-based models for dysgraphia diagnosis, it currently lacks a discussion of existing review/survey papers on this topic. This omission limits the reader's ability to understand how your work builds upon or differs from previous efforts in the field. Include a section that identifies and briefly summarizes existing review papers related to AI-based dysgraphia diagnosis."

While I appreciate the addition of a discussion on previous review papers in Section 3.2, the response remains too general and does not fully address the concern. The discussion broadly mentions limitations of past reviews without explicitly citing and engaging with specific existing surveys on AI-based dysgraphia diagnosis. For example, one recent and comprehensive review (https://link.springer.com/article/10.1007/s10032-024-00464-z) has examined AI methodologies for dysgraphia diagnosis, and provided a structured analysis of automated systems, yet your manuscript does not acknowledge or differentiate itself from this prior work. Instead, it makes broad claims about the shortcomings of earlier reviews without demonstrating how your study adds new insights beyond what has already been explored. To strengthen the scholarly rigor of your manuscript, I recommend explicitly citing and summarizing existing review papers rather than referring to them in a generalized manner, clearly articulating how your review extends, updates, or differs from prior work, and avoiding generalized critiques of previous reviews unless supported by direct comparisons with specific papers. A more detailed engagement with prior surveys will provide a clearer justification for your review and ensure it is properly positioned within the existing body of literature.

Also i suggest changing the title of section 3.2.

Thank you for your insightful comment regarding the clarity of the literature gap our review aims to address. We have revised Section 3.2 to more explicitly highlight how our work fills key gaps left by prior literature. Specifically, unlike earlier reviews that tend to either focus broadly on handwriting difficulties or lean heavily toward clinical perspectives, our review is dedicated exclusively to dysgraphia and places emphasis on artificial intelligence-based diagnostic methods. We address the lack of comprehensive analysis across both traditional machine learning techniques (e.g., SVM, Random Forest) and emerging deep learning architectures (e.g., CNNs).

Additionally, we point out that many earlier reviews overlook modern AI approaches or fail to differentiate between dysgraphia and related learning disabilities. By contrast, our paper aims to provide a focused, up-to-date synthesis of AI-based dysgraphia research. To support this clarification, we have incorporated comparisons with existing reviews and added relevant references where appropriate. These revisions enhance the clarity of our paper's unique contribution to the field.

2. While the added discussion on evaluation metrics (Section 4.1.2) is a valuable improvement, the section would benefit from further expansion on the clinical and practical implications of metric choices. Specifically, it would be useful to discuss how different metrics align with real-world dysgraphia diagnosis scenarios, considering the trade-offs between false positives and false negatives. Additionally, while the discussion acknowledges metric heterogeneity, it would be strengthened by proposing potential future directions for standardization and best practices in evaluating AI models for dysgraphia.

For section 4.1.2, we have significantly expanded this section to incorporate the clinical and practical implications of evaluation metrics in the context of dysgraphia diagnosis. We now clearly explain how different metrics align with real-world use cases. For example, we emphasize that recall (sensitivity) is crucial in school-based screening tools, where the primary goal is to avoid missing at-risk children who may need early intervention. Conversely, precision becomes more critical in clinical settings, where minimizing false positives helps prevent unnecessary diagnoses and the stress or resource strain that may follow.

We also elaborated on the trade-offs between false negatives and false positives, outlining how they can directly affect intervention timing, educational support, and parental response. To address the challenge of metric heterogeneity across studies, we propose that future research adopt a core set of standardized evaluation metrics (e.g., recall, precision, F1-score, AUC), used consistently across tasks and datasets. We also recommend incorporating public benchmark datasets, k-fold cross-validation, and threshold tuning as standard best practices. Lastly, we underscore the value of interdisciplinary collaboration—including clinicians, educators, and AI researchers—in establishing evaluation standards that are both technically robust and practically meaningful. These additions strengthen the section’s relevance and practical utility.

3. While the inclusion of a dedicated section (3.6) on feature analysis is a valuable addition, I believe there is still an opportunity to provide more depth and specificity. The categorization of features into temporal, spatial, pressure-based, and kinematic is a useful framework, but a more detailed discussion of how these features vary in the context of both online and offline handwriting could add significant value. Given the differences between these two modalities, it would be helpful to explore trends in feature selection across studies focusing on each type of handwriting separately. Additionally, discussing the relationship between feature selection and model performance in more detail—perhaps with specific examples from the reviewed studies—would help clarify how different features contribute to model efficacy.

Including relevant citations to support the claims made in this section would also strengthen the discussion, grounding the analysis in the broader body of literature. A brief overview of challenges related to feature standardization across studies would further enrich this section and highlight potential directions for future research. Overall, while the section is a good starting point, expanding on these aspects will provide readers with a more comprehensive understanding of the role that feature selection plays in the development of AI-based dysgraphia diagnostic models.

For your constructive feedback on Section 3.6, we have substantially revised and expanded the section to address each of your suggestions in greater detail.

First, we retained the original categorization of features (temporal, spatial, pressure-based, and kinematic) and developed a more detailed comparison of how these feature types differ between online and offline handwriting modalities. For each feature category, we now provide modality-specific explanations that reflect differences in data acquisition, precision, and diagnostic utility. This includes the limitations of inferring temporal and kinematic features from offline data, as well as the enhanced granularity available from digital tablets in online modalities.

Second, we have improved subsection (3.6.2) discussing feature selection trends across studies, distinguishing between those focused on online handwriting, offline handwriting, and hybrid approaches. This trend analysis draws from specific studies reviewed in our work, with new in-text examples and citations (e.g., [21], [31], [34], [79]) that illustrate which features were selected and how they impacted model performance. For instance, one study achieved a 15% sensitivity improvement by incorporating kinematic features, while another reported high accuracy using spatial features alone in offline datasets. These examples help clarify the performance implications of different feature types, as you suggested.

Third, we have expanded Section 3.6.3 to more thoroughly discuss the relationship between feature selection and model efficacy. We now provide concrete performance outcomes from reviewed studies and explain how certain features align with clinically observable symptoms of dysgraphia (e.g., erratic pressure or prolonged stroke latency), contributing to both diagnostic performance and interpretability.

Fourth, as you recommended, we introduced a brief discussion of feature standardization challenges, including variation in tablet sampling rates, device pressure sensitivity, and inconsistent feature definitions across studies. This lack of standardization presents a barrier to reproducibility and comparability across models and was added to highlight future research needs.

Finally, we ensured that these revisions are supported by relevant citations to anchor our analysis in the broader literature. These updates strengthen Section 3.6 by providing the specificity, depth, and empirical grounding necessary to help readers better understand the pivotal role of feature selection in AI-based dysgraphia diagnosis.

4.The recommendations outlined in future works section provide a solid foundation for advancing dysgraphia detection models. The focus on combining individual fusion models with traditional machine learning approaches, addressing dataset limitations, and leveraging CNNs for improved performance are highly relevant and aligned with current research trends. However, a few aspects could benefit from further elaboration. For example, while the synergy between fusion models and traditional methods is well-highlighted, additional context or examples on how such integration could be practically implemented would strengthen this point. It would also be useful to address potential challenges, such as computational complexity or data requirements, associated with this integration. Regarding dataset limitations, the mention of data augmentation methods is insightful, but a more detailed explanation of how these techniques can be specifically applied to handwriting data in the context of dysgraphia would be beneficial. Additionally, discussing how future studies could ensure that datasets are more diverse in terms of demographics, language, and cultural context would help provide a broader perspective.

Moreover, referencing studies that have successfully employed fusion models, advanced data augmentation, or CNNs in handwriting or related tasks could give a clearer picture of how these methods can be translated into dysgraphia detection. This would also demonstrate that the proposed future work is well-supported by the current body of research, giving it more credibility and a stronger basis for consideration.

We greatly appreciate your thoughtful feedback on the future works section and the encouragement to provide further elaboration. In response, we have revised the section to strengthen the depth and clarity of our recommendations, addressing all key points raised.

To elaborate on the integration of fusion models with traditional machine learning, we now describe how handcrafted or statistical features, extracted from traditional preprocessing pipelines (e.g., kinematic or pressure-based analysis), can be combined with deep feature representations and passed into conventional classifiers such as SVMs or Random Forests. This hybrid approach enables the integration of domain knowledge with learned representations, resulting in more robust and interpretable models. However, we also acknowledge potential challenges, such as increased computational complexity, limited scalability, and the difficulty of optimally weighting heterogeneous feature sources. These trade-offs are now explicitly discussed to give a more balanced and practical outlook.

Regarding dataset limitations and augmentation, we expanded the discussion to clarify how techniques such as shearing, shifting, mirroring, and flipping can simulate natural variations in handwriting, including stroke rhythm, pressure, and alignment—all of which are known indicators of dysgraphia. This specificity helps ground the relevance of data augmentation strategies to dysgraphia-related motor and spatial irregularities.

Additionally, we have addressed the need for greater dataset diversity by emphasizing the importance of including samples from diverse demographics, cultural contexts, and languages. We also expanded on dataset diversity via multi-center collaborations and non-Latin scripts (e.g., Arabic, Devanagari), ensuring inclusivity and reducing bias. This broader inclusion is crucial for enhancing generalizability and mitigating algorithmic bias.

To support these future directions, we have incorporated citations to studies that have successfully implemented fusion strategies, augmentation techniques, and CNN architectures in handwriting and related tasks. These references demonstrate the feasibility and prior success of the methods discussed, reinforcing the credibility of the proposed directions.

5. While Tables 6 to 11 present general advantages and disadvantages of the SVM, KNN, Random Forest algorithms etc..., they would benefit from a more focused discussion contextualized to dysgraphia diagnosis. For example, SVM’s effectiveness with high-dimensional but small handwriting datasets has been reported in [Ref], while KNN’s simplicity makes it a common baseline in early-stage studies despite its sensitivity to noise. Random Forest’s ability to handle mixed feature types (e.g., spatial, temporal, pressure-based) has proven useful in multimodal handwriting analysis [Ref]. Including such application-specific insights would enhance the relevance of these comparisons and help readers understand the trade-offs in algorithm selection for this domain.

Thank you for your valuable feedback. To fully address Comment 5, we have revised Tables 6–11 to include dysgraphia-specific insights in the advantages and disadvantages of each algorithm (SVM, KNN, Random Forest, K-means, AdaBoost, CNN), focusing on their application to handwriting data and dysgraphia diagnosis challenges. Specifically, we:

Updated Table Titles: Changed the titles of Tables 6–11 to explicitly reference dysgraphia diagnosis (e.g., Table 6: “Advantages and Disadvantages of SVM Algorithm in Dysgraphia Diagnosis”) to clarify the context.

Added Dysgraphia-Specific Insights: Incorporated application-specific advantages and disadvantages, such as SVM’s effectiveness with small, high-dimensional handwriting datasets, KNN’s role as a baseline despite noise sensitivity, Random Forest’s ability to handle multimodal features, K-means’ clustering of dysgraphic patterns, AdaBoost’s enhancement of weak classifiers, and CNN’s strength in complex image patterns despite dataset limitations.

Included Trade-off Discussion: Added a new paragraph at the end of Section 3.5 to discuss trade-offs in algorithm selection, highlighting how dataset characteristics (

---

## [Decision Letter · Decision Letter 2]

15 Jun 2025

Dear Dr. Sajedi,

Thank you for submitting your manuscript to PLOS ONE. After careful consideration, we feel that it has merit but does not fully meet PLOS ONE’s publication criteria as it currently stands. Therefore, we invite you to submit a revised version of the manuscript that addresses the points raised during the review process.

We look forward to receiving your revised manuscript.

Kind regards,

Sadiq H. Abdulhussain, Ph.D.

Academic Editor

PLOS ONE

Journal Requirements:

Reviewers' comments:

Reviewer's Responses to Questions

**Comments to the Author**

Reviewer #2: (No Response)

2. Is the manuscript technically sound, and do the data support the conclusions?

Reviewer #2: Yes

3. Has the statistical analysis been performed appropriately and rigorously?

Reviewer #2: N/A

4. Have the authors made all data underlying the findings in their manuscript fully available?

Reviewer #2: Yes

5. Is the manuscript presented in an intelligible fashion and written in standard English?

Reviewer #2: Yes

Reviewer #2: I appreciate your thorough revisions and the improvements made throughout the manuscript. Most of my earlier comments have been addressed satisfactorily.

However, one key issue remains regarding the treatment of related literature in Section 3.2. In your response, you mention prior reviews in a general sense, but you do not cite or engage directly with the recent and highly relevant review by Kunhoth et al. (2024), "Automated systems for diagnosis of dysgraphia in children: a survey and novel framework". This work overlaps closely in scope with your own and offers a comprehensive review of AI-based dysgraphia diagnosis, including both traditional and deep learning approaches, feature-level discussions, and even non-ML tools.

I strongly encourage you to cite this work explicitly and include a short structured comparison that clarifies how your review builds upon or differs from it. This will improve the scholarly positioning and rigor of your manuscript.

Once this addition is made, I believe the manuscript will be in strong shape for publication.

**Do you want your identity to be public for this peer review?** For information about this choice, including consent withdrawal, please see our Privacy Policy

Reviewer #2: No

---

## [Author Response · Author response to Decision Letter 3]

3 Jul 2025

Note: To enhance clarity, we have highlighted the reviewers' comments and used Arial font for them, while our responses are provided in Times New Roman.

Third Revision

Reviewer #2: I appreciate your thorough revisions and the improvements made throughout the manuscript. Most of my earlier comments have been addressed satisfactorily.

However, one key issue remains regarding the treatment of related literature in Section 3.2. In your response, you mention prior reviews in a general sense, but you do not cite or engage directly with the recent and highly relevant review by Kunhoth et al. (2024), "Automated systems for diagnosis of dysgraphia in children: a survey and novel framework". This work overlaps closely in scope with your own and offers a comprehensive review of AI-based dysgraphia diagnosis, including both traditional and deep learning approaches, feature-level discussions, and even non-ML tools.

I strongly encourage you to cite this work explicitly and include a short structured comparison that clarifies how your review builds upon or differs from it. This will improve the scholarly positioning and rigor of your manuscript.

Once this addition is made, I believe the manuscript will be in strong shape for publication.

Our response:

Thank you for highlighting the importance of comparing our work with other comprehensive reviews. In response, we have expanded the discussion in the related work section to include a detailed comparison with the recent review by Kunhoth et al., 2024 [1], which offers valuable insights into learning disorders and predictive methods, both machine learning and non-machine learning based.

We clarify that while Kunhoth et al. provide a broad overview and effective categorization across various disorders, our review focuses specifically on dysgraphia and delivers more in-depth technical analysis, including a detailed breakdown of individual models, feature selection, and implementation strategies. We emphasize how our review contributes beyond general knowledge by offering model-level insights, highlighting recent AI advancements, and synthesizing results from studies involving modern, hybrid approaches like the integration of spatial and dynamic handwriting features.

Before:

“The publication dates span nearly a decade, from 2015 to the present (Fig 3). Of the 21 selected papers, 8 (n = 8, 38%) [34,35,38,42-44,47] are conference papers, and the remaining 13 (n = 13, 62%) [20,31-33,36,39,40,41,45,46,48-57] are journal articles. The papers originate from various countries, including India, Qatar, Spain, Switzerland, Slovakia, the Czech Republic, Israel, France, the USA, Malaysia, Morocco, and Austria. Additionally, some papers are collaborative efforts involving researchers from multiple nationalities. In addition, only nine of the included studies are open-access (n = 9, 42%) [20,31-33,40,41,45,49], and the 12 other papers are private (n = 12, 58%) [34-39,42-44,46-48].

While our review provides detailed insights into the current AI methodologies for diagnosing dysgraphia, it is essential to contextualize this within the broader academic dialogue. Previous reviews have provided foundational knowledge in this field; however, most adopt a clinical perspective, prioritizing diagnostic criteria and behavioral assessments over AI’s potential to transform dysgraphia diagnosis [50,51]. Moreover, few are dedicated solely to dysgraphia, often embedding it within broader discussions of specific learning disabilities (SLDs) like dyslexia and dyscalculia, thus diluting focus on dysgraphia-specific challenges [52-54]. Even reviews exploring AI applications, such as [55,56], typically emphasize traditional machine learning methods (e.g., SVM, AdaBoost) and overlook state-of-the-art approaches, notably the integration of traditional and modern techniques, such as combining handwriting image analysis with dynamic features like pen pressure or motor patterns, which our review examines in detail. Additionally, many prior reviews are outdated, failing to reflect recent AI advancements. Our work addresses these gaps by providing a dysgraphia-specific, up-to-date synthesis of AI applications, comprehensively covering both traditional and cutting-edge methods. This contribution highlights the multidisciplinary relevance of AI in dysgraphia diagnosis through handwritten image analysis, underscoring its potential impact across educational and healthcare settings. Despite growing interest, a recent slowdown in publications suggests a bottleneck in data availability, emphasizing the need for improved data collection and sharing to sustain research momentum in this critical area.”

After:

“The publication dates span nearly a decade, from 2015 to the present (Fig 3). Of the 21 selected papers, 8 (n = 8, 38%) [34,35,38,42-44,47] are conference papers, and the remaining 13 (n = 13, 62%) [20,31-33,36,39,40,41,45,46,48-58] are journal articles. The papers originate from various countries, including India, Qatar, Spain, Switzerland, Slovakia, the Czech Republic, Israel, France, the USA, Malaysia, Morocco, and Austria. Additionally, some papers are collaborative efforts involving researchers from multiple nationalities. In addition, only nine of the included studies are open-access (n = 9, 42%) [20,31-33,40,41,45,49], and the 12 other papers are private (n = 12, 58%) [34-39,42-44,46-48].

While our review provides detailed insights into the current AI methodologies for diagnosing dysgraphia, it is essential to contextualize this within the broader academic dialogue. Previous reviews have provided foundational knowledge in this field; however, most adopt a clinical perspective, prioritizing diagnostic criteria and behavioral assessments over AI’s potential to transform dysgraphia diagnosis [50,51]. Moreover, few are dedicated solely to dysgraphia, often embedding it within broader discussions of specific learning disabilities (SLDs) like dyslexia and dyscalculia, thus diluting focus on dysgraphia-specific challenges [52-54]. Even reviews exploring AI applications, such as [55,56], typically emphasize traditional machine learning methods (e.g., SVM, AdaBoost) and overlook state-of-the-art approaches, notably the integration of traditional and modern techniques, such as combining handwriting image analysis with dynamic features like pen pressure or motor patterns, which our review examines in detail. Additionally, many prior reviews are outdated, failing to reflect recent AI advancements. One of the more comprehensive recent works is the review by Kunhoth et al. (2024) [57], which provides a broad overview of various learning disorders and explores both machine learning and non-machine learning approaches for their prediction. While their work offers valuable general insights and effectively groups and compares different models, it lacks the depth of analysis found in our study. In contrast, our review includes more specific technical details, considers a wider array of models, and delves into how they are implemented. It also provides deeper guidance and recommendations, making it more actionable for researchers and practitioners. Together, these additions make our review not only complementary to previous works but also a necessary update for the current state of AI research in this domain. Our work addresses these gaps by providing a dysgraphia-specific, up-to-date synthesis of AI applications, comprehensively covering both traditional and cutting-edge methods. This contribution highlights the multidisciplinary relevance of AI in dysgraphia diagnosis through handwritten image analysis, underscoring its potential impact across educational and healthcare settings. Despite growing interest, a recent slowdown in publications suggests a bottleneck in data availability, emphasizing the need for improved data collection and sharing to sustain research momentum in this critical area.”

Other Minor Changes:

As part of the future recommendations, we have added an example [2] from our recently published work to demonstrate the practical implementation of integrating fusion models with traditional approaches. This real-world case enhances the clarity and applicability of the proposed direction, offering readers a concrete understanding of how such integration can be effectively realized.

References:

1. Kunhoth, J., Al-Maadeed, S., Kunhoth, S., Akbari, Y., & Saleh, M. (2024). Automated systems for diagnosis of dysgraphia in children: a survey and novel framework. International Journal on Document Analysis and Recognition (IJDAR), 27(4), 707-735.

2. Vakili, Y. Z., Fallah, A., Esmaeili, K., & Mirfazeli, F. S. (2025). Revolutionizing Dysgraphia Detection: Combining Feature Fusion with Non-Discriminatory Regularization. 2025 11th International Conference on Web Research (ICWR), 250–256. https://doi.org/10.1109/ICWR65219.2025.11006233

---

## [Decision Letter · Decision Letter 3]

7 Jul 2025

AI-Driven Approaches for Dysgraphia Diagnosis Using Online and Offline Handwriting Data: A Comprehensive Scoping Review

PONE-D-24-28058R3

Dear Dr. Sajedi,

We’re pleased to inform you that your manuscript has been judged scientifically suitable for publication and will be formally accepted for publication once it meets all outstanding technical requirements.

Kind regards,

Sadiq H. Abdulhussain, Ph.D.

Academic Editor

PLOS ONE

Additional Editor Comments (optional):

Reviewers' comments:

Reviewer's Responses to Questions

**Comments to the Author**

Reviewer #2: All comments have been addressed

2. Is the manuscript technically sound, and do the data support the conclusions?

Reviewer #2: Yes

3. Has the statistical analysis been performed appropriately and rigorously?

Reviewer #2: N/A

4. Have the authors made all data underlying the findings in their manuscript fully available?

Reviewer #2: Yes

5. Is the manuscript presented in an intelligible fashion and written in standard English?

Reviewer #2: Yes

Reviewer #2: All comments have been addressed.

**Do you want your identity to be public for this peer review?** For information about this choice, including consent withdrawal, please see our Privacy Policy

Reviewer #2: No

---

## [Editor Report · Acceptance letter]

PONE-D-24-28058R3

PLOS ONE

Dear Dr. Sajedi,

I'm pleased to inform you that your manuscript has been deemed suitable for publication in PLOS ONE. Congratulations! Your manuscript is now being handed over to our production team.

Kind regards,

on behalf of

Dr. Sadiq H. Abdulhussain

Academic Editor

PLOS ONE